# Changing impacts of Alaska-Aleutian subduction zone tsunamis in California under future sea-level rise

Tina Dura [1,2 ✉], Andra J. Garner [3], Robert Weiss[1,2], Robert E. Kopp [4,5], Simon E. Engelhart [6], Robert C. Witter [7], Richard W. Briggs [8], Charles S. Mueller[8], Alan R. Nelson [8] & Benjamin P. Horton [9,10]

The amplification of coastal hazards such as distant-source tsunamis under future relative sea-level rise (RSLR) is poorly constrained. In southern California, the Alaska-Aleutian subduction zone has been identified as an earthquake source region of particular concern for a worst-case scenario distant-source tsunami. Here, we explore how RSLR over the next century will influence future maximum nearshore tsunami heights (MNTH) at the Ports of Los Angeles and Long Beach. Earthquake and tsunami modeling combined with local probabilistic RSLR projections show the increased potential for more frequent, relatively low magnitude earthquakes to produce distant-source tsunamis that exceed historically observed MNTH. By 2100, under RSLR projections for a high-emissions representative concentration pathway (RCP8.5), the earthquake magnitude required to produce >1 m MNTH falls from ~$M_w$9.1 (required today) to $M_w$8.0, a magnitude that is ~6.7 times more frequent along the Alaska-Aleutian subduction zone.

[1] Department of Geosciences, Virginia Tech, Blacksburg, VA 24061, USA. [2] Center for Coastal Studies, Virginia Tech, Blacksburg, VA 24061, USA. [3] Department of Environmental Science, Rowan University, Glassboro, NJ 08028, USA. [4] Department of Earth and Planetary Sciences, Rutgers University, Piscataway, NJ 08854, USA. [5] Institute of Earth, Ocean, and Atmospheric Sciences, Rutgers University, New Brunswick, NJ 08901, USA. [6] Department of Geography, Durham University, Lower Mountjoy, South Road, Durham DH1 3LE, UK. [7] U.S. Geological Survey, Alaska Science Center, Anchorage, AK 99508, USA. [8] U.S. Geological Survey, Geologic Hazards Science Center, Golden, CO 80401, USA. [9] Earth Observatory of Singapore, Nanyang Technological University, Singapore 639798, Singapore. [10] Asian School of the Environment, Nanyang Technological University, Singapore 639798, Singapore. ✉email: tinadura@vt.edu

To the more than one billion people worldwide living in the coastal zone below 10 m of elevation[1], the compound effects of relative sea-level rise (RSLR), tidal flooding[2], and storm surges[3–6] are of increasing concern. Increasing tropical cyclone-driven flood heights due to changes in storm characteristics and RSLR over the past millennium and forecast increasing flood heights in the coming decades have been documented for specific locations such as New York City[7–10]. Other research shows increased extreme sea levels during coastal storms due to RSLR along coasts in California[6] as well as the contiguous United States[11]. Still, other studies have investigated changes to extreme sea levels associated with coastal storms and RSLR in locations such as Australia[12] and Europe[13] and from a global perspective[14]. However, the impacts of RSLR on coastal inundation during other potentially damaging events, such as tsunamis, need further research[15,16]. Along coastlines affected by distant-source tsunamis, where potential tsunami amplitudes are generally on the order of or lower than projected twenty-first-century RSLR, rising baseline sea levels can significantly increase tsunami impacts[16,17].

The economic impacts of future distant-source tsunamis are estimated to be in the billions of dollars for parts of the Pacific Rim, including the southern California coast[18]. Over the past two centuries, at least 14 distant-source tsunamis have damaged the California coast, with 8 of the 14 occurring in the last ~70 years[19]. The economic impacts of distant-source tsunamis in California have increased as coastal populations and infrastructure have grown. At the low-lying, densely populated, and economically important Ports of Los Angeles and Long Beach, recent maximum tsunami amplitudes (defined as the absolute value of the difference between a particular peak or trough of the tsunami and the undisturbed sea level at the time) recorded at tide gauge 1[20] (TG1; Fig. 1c) are associated with distant-source tsunamis from earthquakes originating in Chile (1960; $M_w$9.5; maximum tsunami amplitude = 0.72 m), Alaska (1964; $M_w$9.2; maximum tsunami amplitude = 0.56 m), and Japan (2011; $M_w$9.0; maximum tsunami amplitude = 0.31 m)[19]. The 1960, 1964, and 2011 tsunamis coincided with differing stages of low tide at the ports; therefore, their maximum nearshore tsunami heights (MNTH; defined here as the maximum amplitude of the tsunami wave time series relative to mean sea level (MSL)) were dampened. Nevertheless, strong currents and inundation caused by MNTH of 0.27 m in 1960 and 0.41 m in 1964 caused ~$17 M and ~$149 M (amounts referenced in this paper are in 2019 inflation-adjusted U.S. dollars) of damage in Los Angeles, respectively[19,21]. In 2011, MNTH of 0.13 m created strong currents at the ports and disrupted operations, and higher MNTH (>1 m) along the northern California coast caused ~$115 M in damage[19,21].

Of the distant earthquake source regions posing a tsunami threat to the California coast, the Alaska-Aleutian subduction zone has the potential to produce the highest (1–2 m) tsunami amplitudes[21–23]. The Semidi section of the Alaska-Aleutian subduction zone, defined as the portion of the subduction zone between western Kodiak Island and the Shumagin Islands, has been identified as a source area of particular concern because the continental slope azimuth there directs tsunamis towards southern California (Fig. 1a)[24]. However, historical[25,26] and paleoseismic records[27–29] show that earthquake ruptures in the region are not always confined to the Semidi section[30] and that a rupture in 1788 (M8+), and probably older prehistoric ruptures, propagated east into the Kodiak section (Fig. 1a). Such evidence highlights the need for including multisection (>M9) as well as single-section ruptures along the Semidi and Kodiak sections in southern California and Pacific-wide tsunami hazard assessments[27].

In this work, we explore the effect of RSLR on low-probability, high-impact distant-source tsunami events at the Ports of Los Angeles and Long Beach. We combine single and multi-section earthquake and tsunami modeling along the Alaska-Aleutian subduction zone with probabilistic local RSLR projections for the ports to estimate potential MNTH for the ports during the twenty-first century.

## Results

### Distant-source tsunami simulations for the ports of Los Angeles and Long Beach.
We use a time-independent, deterministic earthquake modeling approach to generate a dataset of possible tsunamigenic earthquake sources ranging from $M_w$8.0 to $M_w$9.4 (15 magnitude steps) along the Semidi and Kodiak sections of the Alaska-Aleutian subduction zone[31]. To account for source variability, we use 50 randomized earthquake areas (composed of adjacent NOAA unit sources) and hypocenter locations (assumed to be in the center of the earthquake area) for each step in earthquake magnitude (Fig. 1b and "Methods" section)[32]. Slip is uniform across unit sources for each earthquake; however, the area (i.e., number of unit sources) over which the slip is distributed for each earthquake magnitude varies (see "Methods" section). Our approach differs from probabilistic tsunami hazard assessments in that we use the same source region, suite of earthquake magnitudes, and slip variations for every year from 2000 to 2100, without considering the probability of each earthquake magnitude. This approach allows us to consider the changing impact of the same suite of significant earthquakes and tsunamis during twenty-first-century RSLR.

Our earthquake modeling (15 earthquake magnitudes × 50 earthquakes per magnitude step) generates 750 deformation fields that are input into GeoClaw[33] for the simulation of tsunami wave propagation (Fig. 1b, Supplementary Table S1 and "Methods" section). We report the resultant distribution of MNTH at the Ports of Los Angeles and Long Beach from our tsunami simulations as the maximum tsunami amplitude at our synthetic tide gauge 2 (TG2; located in the outer harbor at 17 m water depth) relative to MSL (Fig. 1c, d)[20].

We performed a parameter study to see how the MNTH produced by our method of varying the number of unit sources over which slip is distributed for each earthquake magnitude compared to a uniform and non-uniform slip approach for the same earthquake magnitudes (Supplementary Fig. S3). The parameter study supports the conclusion that uniform slip (using a constant number of unit sources and slip for each magnitude while only varying the earthquake location) results in a significant underestimation of MNTH[34,35]. Our method produces a broader range of MNTH similar to a non-uniform slip approach, although there may be an underestimation of the higher extremes of possible MNTH[34,35] (Supplementary Fig. S3). Therefore, the MNTH described in this paper provide a conservative estimate of possible tsunami impacts at the ports. We also performed a parameter study to evaluate how many earthquake areas per magnitude were necessary to produce a consistent and reproducible distribution of MNTH at the ports. Simulating a consistent distribution of MNTH is critical to ensuring that source variability is not the driving factor in changing MNTH. We show that 50 random earthquake areas and hypocenter locations per magnitude step are sufficient to produce a robust statistical representation (i.e., consistent and reproducible) distribution of MNTH at the ports (Supplementary Fig. S4 and "Methods" section).

Note that we do not analyze inundation[15] or currents[36,37], nor do we explore the nonlinear interaction of tides[38] with tsunamis. These processes are complex in a port setting[22] and are computationally prohibitive due to the high-resolution topography and bathymetry needed to obtain reliable results (see "Methods" section). Instead, we use TG2 as a reference point to

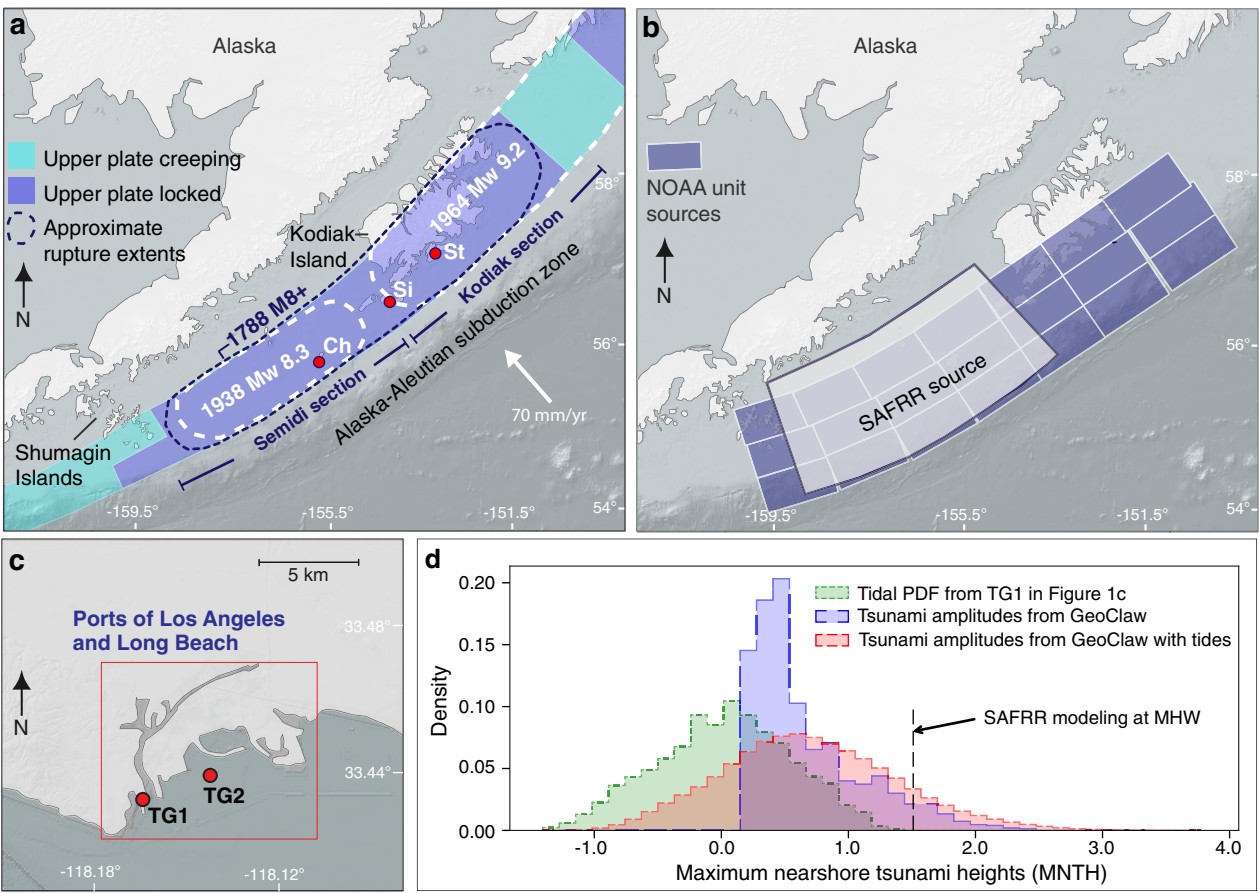

**Fig. 1 Alaska-Aleutian subduction zone tectonic setting and distant-source tsunami modeling. a** Plate tectonic setting of Alaska showing the locked and creeping sections of the Alaska-Aleutian subduction zone, earthquake section boundaries, and approximate historical earthquake extents. Red circles show sites with paleoseismic evidence supporting multi-section earthquake ruptures. Ch Chirikof Island, Si Sitkinak Island, St Sitkalidak Island. **b** Light gray shaded area shows the approximate extent of slip used in the U.S. Geological Survey (USGS) Science Application for Risk Reduction (SAFRR) scenario magnitude 9.1 Semidi section earthquake underlain by a grid of the NOAA unit sources used in this paper. **c** Map of the Ports of Los Angeles and Long Beach showing the location of the long-term tide gauge (est. 1923) measuring water levels at the ports (TG1[67]) and the synthetic tide gauge (TG2) where maximum nearshore tsunami heights (MNTH) were measured in this study. **d** Plot showing the probability density function (PDF) of the MNTH from our suite of modeled earthquake magnitudes in the year 2000 with no tidal variability included (blue histogram), the PDF of the tidal variability at TG1 (green histogram), and the combined MNTH and tidal variability PDF (red histogram). The dashed line shows the 1-m amplitude SAFRR scenario tsunami striking at high tide (MHW)[16], resulting in a MNTH of 1.5 m at the ports.

compare how MNTH distributions are influenced by RSLR over time. We acknowledge that strong and erratic currents may be induced by distant-source tsunamis at the ports (e.g., refs. [22,36]) and that the interaction of tidal currents with distant-source tsunamis may amplify MNTH[38]. Not including these processes in our modeling may underestimate the possible effects of distant-source tsunamis at the ports.

Although not included in our modeling or calculations, we analyzed the earthquake catalog along the entire Alaska-Aleutian subduction zone (see "Methods" section) in order to put the relative frequency of our chosen earthquake magnitudes into context. We used a declustered earthquake catalog ($b$-value = 0.75) to calculate the relative rate of earthquake magnitudes along the Alaska-Aleutian subduction zone (Supplementary Table S2)[39,40]. This approach may underestimate larger magnitude events[41,42]; however, because the catalog at the Alaska-Aleutian subduction zone spans M5-9.2, we believe this is the most conservative approach for calculating relative earthquake rates.

According to our simulations, for the year 2000, without considering RSLR or tidal variability, the 95% central range (CR) of MNTH at the Ports of Los Angeles and Long Beach generated

by tsunamis from $M_w$8.0, $M_w$8.5, and $M_w$9.1 earthquakes is 0.06–0.26 m, 0.12–0.45 m, and 0.33–1.44 m (Supplementary Fig. S1), respectively. For our full suite of earthquakes ($M_w$8.0-$M_w$9.4), the 95% CR of MNTH is 0.19–1.78 m (median MNTH = 0.54 m; Fig. 1d).

The tidal stage during which a tsunami strikes the coast can be very important in determining the highest water levels reached during the event[43–45]. To account for the influence that tidal stage has on MNTH at the Ports of Los Angeles and Long Beach, we linearly combine our MNTH with tidal stages derived from a distribution composed of ~30 years of tidal data from TG1[20] (Fig. 1d and "Methods" section). The maximum tidal range (MLLW-MHHW) observed at the ports is 1.68 m[20]. The resulting MNTH at the ports include rare high and low tsunami height values reflecting the potential for smaller magnitude earthquakes to produce relatively high tsunamis, and larger magnitude earthquakes to produce relatively low tsunamis, due to source and tidal variability (Fig. 1d). Incorporating tidal variability in the calculation of MNTH at the ports results in a 95% CR of MNTH, generated by our full suite of earthquakes ($M_w$8.0–$M_w$9.4), of −0.45–2.03 m (median MNTH = 0.62 m; Fig. 1d).

**Relative sea-level rise at the ports of Los Angeles and Long Beach.** Changes in relative sea level during the twenty-first century vary by location as a result of processes such as atmosphere-ocean dynamics, the gravitational, rotational, and dynamic effects of ocean/cryosphere/hydrosphere mass redistribution, glacio-isostatic adjustment, sediment compaction, and tectonic uplift or subsidence[46,47]. In southern California, gravitational effects are augmented by large contributions from the West Antarctic Ice Sheet (WAIS)[48], considered to be the most vulnerable ice sheet in a warming climate[49], causing the coast to be exceptionally sensitive to RSLR.

To estimate the contribution of RSLR to future MNTH at the Ports of Los Angeles and Long Beach, we use probabilistic projections of local RSLR[50] (see "Methods" section). We examine low and high greenhouse-gas emissions pathways (Representative Concentration Pathways [RCPs] 2.6 and 8.5, respectively). RCP2.6 represents a low-emissions future with mitigation measures most consistent with the Paris Climate Agreement, while RCP8.5 represents a high-emissions future with no firm mitigation targets[51,52]. We consider each pathway under two different treatments of Antarctic Ice Sheet uncertainty (denoted K14 and DP16) (Fig. 2 and Table 1). The K14 projections[50] are consistent with projections from the Intergovernmental Panel on Climate Change (IPCC) Fifth Assessment Report[50,53]. The DP16 projections use outputs from a continental-scale model incorporating marine ice-sheet and ice-cliff instability mechanisms and are highly sensitive to atmospheric warming[54,55]. Using the year 2000 as a baseline, we generate projections of local RSLR at the Ports of Los Angeles and Long Beach for the two emissions pathways during each decade of the twenty-first century.

**Future maximum nearshore Tsunami heights.** To study the influence of future RSLR on MNTH at the Ports of Los Angeles and Long Beach over the twenty-first century, we linearly combined a subsample of local probabilistic RSLR projections (RCP2.6 K14 and DP16, RCP8.5 K14 and DP16) spanning the low and high-end values of RSLR for each decade with our MNTH distribution including tidal variability. Our approach of linearly combining MNTH with RSLR projections resulted in a difference of −7% to +15% in MNTH compared to an approach that accounts for changing bathymetry due to RSLR for each tsunami simulation (see "Methods" section). Previous tsunami[15] and storm surge[10] studies also highlight the potential nonlinear

effect of RSLR on flood heights over time, but the effect is small for RSLR of < 2m[56].

At the ports, RSLR causes tsunamis from our full suite of earthquakes to have higher median MNTH in 2050, 2070, and 2100 than today (Fig. 3 and Supplementary Table S3). The median of the MNTH distribution with tidal variability in 2000 is 0.62 m, reflecting the median of distant-source tsunamis generated by all earthquakes ($M_w$8.0 to $M_w$9.4) with no RSLR included. By 2050, median MNTH distributions range from 0.74 m (RCP2.6 DP16) to 0.84 m (RCP8.5 K14). By 2070, the median of MNTH distributions ranges from 0.83 m for RCP2.6 DP16 to 1.08 m for RCP8.5 DP16. The medians of the MNTH distributions in 2100 reflect the wide range in emissions pathways and sea-level projections, ranging from 0.95 m for RCP2.6 DP16 to 1.86 m for RCP8.5 DP16.

To explore the temporal relationship between earthquake magnitude and MNTH, we calculated the earthquake magnitude that generates tsunamis that have a 50% chance of exceeding a defined MNTH as a function of time (Fig. 4 and Table 2). We examined three MNTH: 0.5 m (measured multiple times during the historical period), 1.0 m (2 times larger than any historical event), and 1.5 m (similar to a previous tsunami scenario conducted at the ports[16] and the highest storm-driven extreme sea level recorded at the ports[57]).

For all RSLR projections considered (RCP2.6 K14 and DP16, RCP8.5 K14 and DP16), the earthquake magnitude required to exceed MNTH of 0.5 m drops to the lowest considered magnitude between 2040 and 2060. Today, a ∼$M_w$8.7 earthquake generates tsunamis that have a 50% chance of exceeding MNTH of 0.5 m.

**Table 1 RSLR projections (in meters) above 1991–2009 mean sea level for the years 2050, 2070, and 2100 at the Ports of Los Angeles and Long Beach.**

|  | 2050 | 2070 | 2100 |
|---|---|---|---|
| RCP 2.6 K14 | 0.07–0.33 | 0.10–0.54 | 0.12–0.95 |
| RCP 2.6 DP16 | −0.01–0.28 | 0.01–0.45 | 0.03–0.73 |
| RCP 8.5 K14 | 0.10–0.37 | 0.18–0.67 | 0.28–1.29 |
| RCP 8.5 DP16 | 0.04–0.37 | 0.19–0.94 | 0.57–2.4 |

RSLR projections are 95% credible intervals for the RCP2.6 and RCP8.5 emissions pathways under two treatments of Antarctic Ice Sheet uncertainty (K14 and DP16)[60,68].

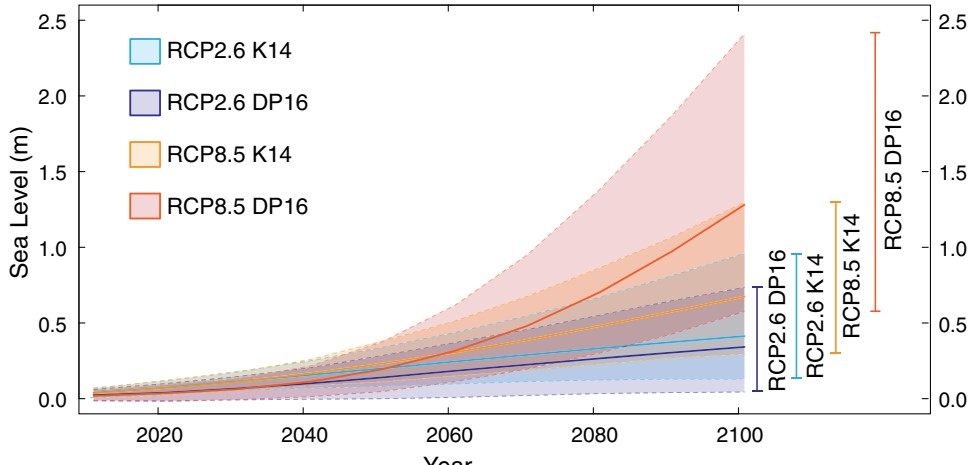

**Fig. 2 Local probabilistic sea-level projections for the Ports of Los Angeles and Long Beach from 2000 to 2100.** Projections are calculated using RCP2.6 (blue) and RCP8.5 (orange) projections[50] and for projections combining enhanced Antarctic Ice Sheet (AIS) contributions from[54] with the RCP2.6 (purple) and RCP8.5 (red) projections from (DP16[60]). Solid lines in the shaded areas show the median of each projection and dashed lines bordering the shaded area and vertical lines on the right show the 95% credible intervals.

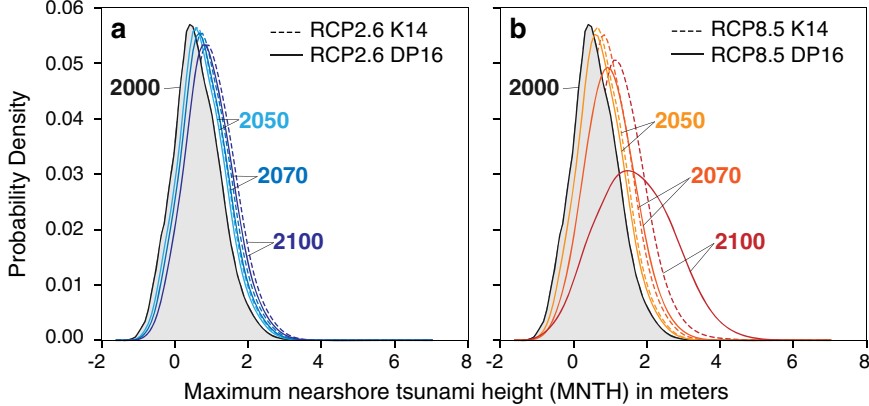

**Fig. 3 Normalized distributions of maximum nearshore tsunami heights (MNTH) at the Ports of Los Angeles and Long Beach.** The MNTH distribution for 2000 reflects the range of possible modeled MNTH, including tidal variability and does not include sea-level rise. MNTH distributions for 2050, 2070, and 2100 are calculated by combining the same range of modeled MNTH with **a** low-emissions (RCP2.6) sea-level projections with the K14[68] Antarctic Ice Sheet (AIS) contribution and DP16[60] AIS contribution. **b** high-emissions (RCP8.5) sea-level projections with K14[68] AIS contribution and DP16[60] AIS contribution.

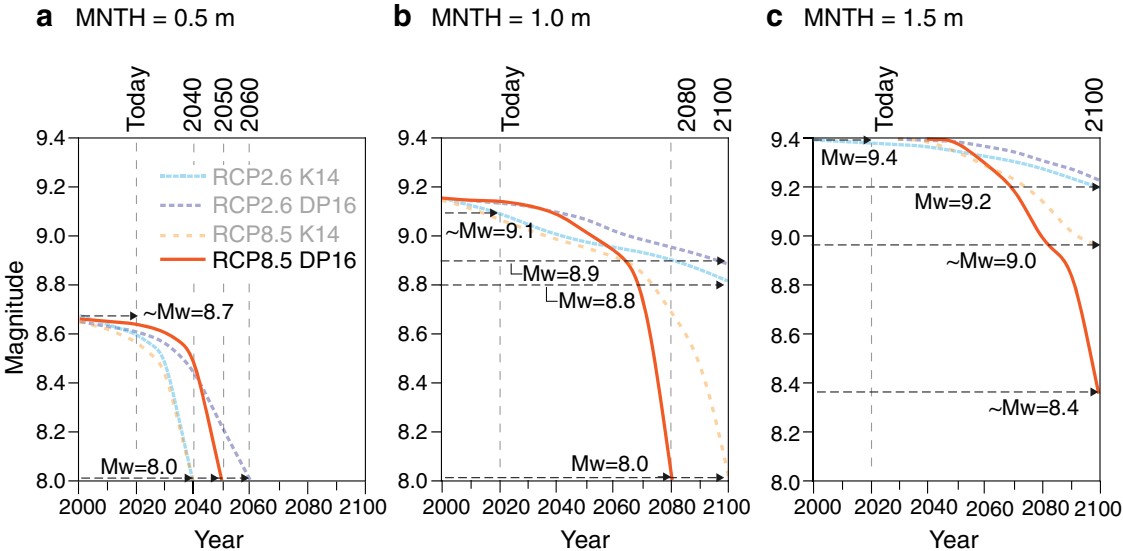

**Fig. 4 Plots showing the earthquake magnitude that produces tsunamis that have a 50% chance of exceeding defined maximum nearshore tsunami heights (MNTH) as a function of time.** We examined three MNTH: **a** 0.5 m (measured multiple times during the historical period), **b** 1.0 m (two times larger than any historical event), and **c** 1.5 m (similar to a previous tsunami scenario conducted at the ports[16] and the highest storm-driven extreme sea level recorded at the ports[57]).

**Table 2 The earthquake magnitude that has a 50% chance of generating a tsunami that exceeds flood heights of 0.5, 1.0, and 1.5 m today (2020) and in the year 2040\*, 2050\*\*, or 2060\*\*\* (for 0.5 m flood heights) or 2100 (for 1.0 m MNTH and 1.5 m MNTH) under the four RSLR projections considered.**

|  | 0.5 m | 0.5 m | 1.0 m | 1.0 m | 1.5 m | 1.5 m |
|  | 2020 | 2040–60 | 2020 | 2100 | 2020 | 2100 |
|---|---|---|---|---|---|---|
| RCP 2.6 K14 | ~$M_w$8.7 | $M_w$8.0* | ~$M_w$9.1 | $M_w$8.8 | $M_w$9.4 | $M_w$9.2 |
| RCP 2.6 DP16 | ~$M_w$8.7 | $M_w$8.0*** | ~$M_w$9.1 | $M_w$8.9 | $M_w$9.4 | $M_w$9.2 |
| RCP 8.5 K14 | ~$M_w$8.7 | $M_w$8.0* | ~$M_w$9.1 | $M_w$8.0 | $M_w$9.4 | ~$M_w$9.0 |
| RCP 8.5 DP16 | ~$M_w$8.7 | $M_w$8.0** | ~$M_w$9.1 | $M_w$8.0 | $M_w$9.4 | ~$M_w$8.4 |

By 2040 (RCP2.6 K14 and RCP8.5 K14), 2050 (RCP8.5 DP16), and 2060 (RCP2.6 DP16), a $M_w$8.0 or lower earthquake generates tsunamis that have the same probability of exceeding MNTH of 0.5 m (Fig. 4a). Along the Alaska-Aleutian subduction zone, a $M_w$8.0 earthquake is ~3.4 times more likely to occur than a ~$M_w$8.7 (see "Methods" section).

The influence of different RSLR projections becomes more apparent for higher MNTH. For RCP2.6 K14 and DP16, there is a small decrease in the earthquake magnitude required to exceed MNTH of 1.0 m by 2100; today, a ~$M_w$9.1 earthquake generates tsunamis that have a 50% chance of exceeding MNTH of 1.0 m. By 2100, an earthquake of $M_w$8.9 (RCP2.6 DP16) or $M_w$8.8

(RCP2.6 K14) generates tsunamis that have the same probability of exceeding MNTH of 1.0 m (Fig. 4b). A ~$M_w$8.8 earthquake is ~1.7 times more likely to occur than a ~$M_w$9.1 along the Alaska-Aleutian subduction zone. For RCP8.5, we see a significant decrease in the earthquake magnitude—$M_w$8.0—that has a 50% chance of exceeding MNTH of 1.0 m by 2080 (DP16) and 2100 (K14). A $M_w$8.0 earthquake is ~6.7 times more likely to occur than a ~$M_w$9.1 along the Alaska-Aleutian subduction zone.

The wide range of RSLR projections strongly influences future MNTH of 1.5 m or higher (Fig. 4c). For RCP2.6 K14 and DP16, we see a small decrease in the earthquake magnitude required to exceed MNTH of 1.5 m by 2100; today, a $M_w$9.4 or larger earthquake generates tsunamis that have a 50% chance of exceeding MNTH of 1.5 m. By 2100, under RCP2.6 RSLR, an earthquake of approximately $M_w$9.2 generates tsunamis that have the same probability of exceeding MNTH of 1.5 m. For RCP8.5 K14, a ~$M_w$9.0 earthquake generates tsunamis that have a 50% chance of exceeding MNTH of 1.5 m by 2100. A ~$M_w$9.0 earthquake is 2.0 times more likely to occur than a ~$M_w$9.4 along the Alaska-Aleutian subduction zone. For RCP8.5 DP16, we see a more rapid and substantial drop in the earthquake magnitude required to produce MNTH of 1.5 m: by 2100, a ~$M_w$8.4 earthquake generates tsunamis that have a 50% chance of exceeding MNTH of 1.5 m. A $M_w$8.4 earthquake is ~5.6 times more likely to occur than a ~$M_w$9.4 along the Alaska-Aleutian subduction zone (see "Methods" section).

## Discussion

The U.S. Geological Survey (USGS) Science Application for Risk Reduction (SAFRR) project modeled a $M_w$9.1 earthquake and its tsunami sourced along the Semidi section of the Alaska-Aleutian subduction zone (Fig. 1b). The SAFRR tsunami scenario found that such an earthquake could produce a distant-source tsunami with an amplitude of ~1 m at the Ports of Los Angeles and Long Beach. In the scenario, the tsunami struck the coast during mean high water (MHW), producing MNTH of 1.59 m, and causing losses of up to ~$4.2 billion (Fig. 1d)[18]. However, the SAFRR tsunami scenario did not consider the amplification of MNTH under RSLR over the next century. In addition, the scenario did not consider multi-section, higher magnitude earthquake ruptures creating higher tsunami amplitudes or the varying tidal stages during which a tsunami may strike. Both may dampen or amplify MNTH.

By including future RSLR, multi-section ruptures, and tidal variability in our tsunami modeling, we provide a more complete picture of potential future MNTH at the Ports of Los Angeles and Long Beach. We show that under rising sea levels, the possibility of economically and socially disruptive distant-source tsunami events like the one simulated in the SAFRR scenario[18] will increase as the earthquake magnitude required to exceed MNTH of >1.0 m drops dramatically (from $M_w$9.1 to a $M_w$8.0). A similar increase in flood frequencies under future RSLR in southern California has been predicted in storm surge and tidal flooding studies[6,11].

Our results highlight the need to consider RSLR when assessing possible future MNTH and planning for RSLR at the Ports of Los Angeles and Long Beach. Finally, given that changing sea levels threaten coastal communities around the globe, these results suggest that RSLR should also be considered as part of the planning and decision-making process at other distant- and near-source tsunami-prone coasts worldwide[17].

## Methods

**Probabilistic sea-level rise projections**. In the coming centuries, global mean sea level (GMSL) will continue to rise due to the warming climate, generating hazards for coastal populations, economies, infrastructure, and ecosystems around the

world[58]. For a low-emissions future with mitigation measures most consistent with the Paris Climate Agreement [Representative Concentration Pathway (RCP) 2.6[51,52]], the Fifth Assessment Report (AR5) from the Intergovernmental Panel on Climate Change (IPCC) projected a "likely" (> 66% probability) GMSL rise of 0.28–0.61 m by 2100 relative to 1986–2015[53]. For a high-emissions "business as usual" future with no particular mitigation targets (RCP 8.5), IPCC AR5 projected a "likely" GMSL rise of 0.52–0.98 m by 2100 relative to 1986–2015[53]. However, recent studies suggest the IPCC AR5 GMSL rise estimates may be too low. For example, a physical ice-sheet model that includes important mechanisms such as ice-shelf hydrofracturing and the structural collapse of marine-terminating ice-cliffs shows the Antarctic Ice Sheet (AIS) could contribute far greater amounts to GMSL rise than previously expected: >1.5 by 2100 under RCP8.5[59,60]. The possibility of higher-end projections being realized or even exceeded has many stakeholders preparing for sea-level rise in excess of 1 m by 2100[61], and under extreme scenarios 2+ m of sea-level rise by 2100[62].

The early impacts of higher sea levels are already being felt in southern California, where coastal flooding during storms, periodic tidal flooding, and increased coastal erosion are becoming more frequent and extensive[63–65]. Higher sea levels will also leave southern California coastlines even more vulnerable to distant-source tsunamis, which have repeatedly impacted the coast in the last few centuries[66].

We consider probabilistic sea-level rise projections from the Los Angeles tide gauge (33° 43.2′ N, 118° 16.4′ W[67]; TG1 in Fig. 1c)[60] under two methodological approaches for the Antarctic ice sheet (K14 and DP16) and two different emissions pathways (RCP2.6 and RCP 8.5)[60,68]. RSL projections are determined from 10,000 Monte Carlo samples of SLR projected at the Los Angeles Tide Gauge using the approach outlined in Kopp et al.[68]. Both K14 and DP16 projections combine estimates of thermal expansion and ocean dynamics from CMIP5 global climate model simulations, glacier melt from CMIP5-forced surface-mass balance modeling, global land water storage changes from semi-empirical modeling, long-term non-climatic relative sea-level change based on spatio-temporal statistical modeling of tide-gauge data, and gravitational, rotational, and deformational effects of glacier and ice sheet changes from geophysical modeling. Both sets also use Greenland ice sheet projections based on a combination of AR5 expert assessment regarding likely changes and tail-risk information from the structured expert judgment exercise of Bamber and Aspinall[69]. The K14 projections use the same approach for Antarctica and are thus consistent with the projections of AR5)[53,68], while the DP16 projections[60] replaced the K14 Antarctic projection with projections from the continental-scale ice-sheet/ice-shelf model of Deconto and Pollard[59]. The DP16 projections were the first continental-scale RCP-driven projections to account for marine ice-cliff instability associated with the combined effects of ice-shelf hydrofracturing and gravitational instability[59,60,70]. The two sets of projections taken together provide a reasonable approach to bracketing the range of plausible probability distributions of future rise[47].

**Earthquake realizations**. To study the influence of future relative sea-level rise (RSLR) on tsunami impacts in the Ports of Los Angeles and Long Beach, we consider the same suite of earthquakes between $M_w$8.0 and $M_w$9.4 (with 15 magnitude steps) in each year. We do not take the occurrence probability of each earthquake magnitude into account in order to focus on the tsunami impact that each magnitude step would have under different sea-level rise projections. For our analysis, we generate robust distributions of maximum nearshore tsunami heights (MNTH) that are based on 50 spatially varying earthquakes in each magnitude step to address uncertainties in epicenter locations (see more details on this approach in the paragraph below). We also considered another possible source of uncertainty in the MNTH generated by our earthquake realizations: uniform vs. non-uniform slip. Traditionally, it was assumed that the tsunami-wave dynamics filters and averages the slip differences across the rupture area as the tsunami propagates away from the source area. However, Li et al. and Melgar et al.[15,34,71] demonstrated that tsunami amplitudes are underestimated in earthquakes with uniform slip across the earthquake area, even in the far-field. Indeed, slip inversion studies of historical earthquakes show that slip distribution varies greatly across the plate interface, and if this is not considered in earthquake realizations, MNTH will be underestimated[35]. To avoid underestimating tsunami amplitude in the Ports of Los Angeles and Long Beach, we devised the method of varying the number of unit sources over which slip is distributed for each earthquake magnitude (described below). Our approach is similar to the variable-area-uniform-slip (VAUS) applied by Davies[35].

We employ the empirical equations from Strasser[72] to estimate the average earthquake area that characterizes each magnitude step (15 magnitude steps between $M_w$8.0 and $M_w$9.4). Because the earthquake area varies in each magnitude step around the average, we multiply the calculated earthquake area by a random factor varying between 0.5 and 1.5 to produce 50 randomized earthquake areas (15 magnitude steps × 50 earthquake areas per magnitude = 750 simulations). We use the 50 earthquake areas per magnitude step to determine how many adjacent NOAA unit sources (100 km × 50 km)[32] per earthquake have to be used to cover the earthquake area. At the same time, to vary the hypocenter locations within each magnitude step, we place the unit sources associated with each earthquake randomly within the unit source grid of the Semidi and Kodiak sections (Fig. 1b).

Unit sources are not superposed. It should be noted that the arrangement of the unit sources is loosely constrained by the length of the computed earthquake area to avoid unrealistically long and thin earthquake areas. To determine the slip ($D$; in this study, slip is uniform across unit sources) for each of the 50 randomized earthquake areas and hypocenter locations per magnitude step, we use the following equation, derived from the moment magnitude and the seismic moment:

$$D = \frac{10^{1.5(M_w + 10.7)}}{\mu A}$$

For example, a $M_w 8.1$ can be composed of one, two, or three-unit sources, translating into slips of $D \approx 8$ m (one unit source), $D \approx 4$ m (two unit sources), or $D \approx 2.7$ m (three unit sources). Thus, for each earthquake magnitude, slip can be concentrated over a variable amount of unit sources to account for slip variability within each magnitude step. The minimum, average, and maximum slip and resultant deformation for each magnitude step are reported in Supplementary Table S1.

**Tsunami simulations**. To generate the initial conditions for the tsunami simulations from the aforementioned suite of earthquakes, we employ the Okada deterministic method[31] to generate the maximum vertical ocean-surface deformation that is responsible for tsunami generation (Supplementary Table S1). Okada's method requires the surface area of the earthquake, its depth, rake, strike, and dip. We utilize the NOAA tsunami unit sources to provide the required information (depth: 5–40 km, rake: 90°, strike: see unit source orientation on Fig. 1b, dip: 15°[32]).

We use GeoClaw to carry out the tsunami simulations. GeoClaw is a widely used modeling tool for geophysical flows and is part of ClawPack software package[73]. GeoClaw has been validated and verified[74] and applied to a variety of different past tsunamis, such as the 2015 Chile[75] and 2011 Japan[76,77] events. One of the main advantages of GeoClaw is the Adaptive Mesh Refinement (AMR) that is implemented and available for tsunami simulations. AMR automatically refines the computational grid in areas where a finer resolution of the employed grid helps to find a more accurate numerical solution. For tsunami simulations, this area coincides with the traveling tsunami wave train. It is possible to force the refinement level in certain areas (e.g., southern California), which is a powerful method to reduce the computational time, for example, when many model runs are needed such as in our case. Supplementary Fig. S2 depicts the refinement areas we used in our simulations. As mentioned earlier, GeoClaw employs AMR to make the computations more efficient. In the open Pacific Ocean (Supplementary Fig. S2A, Box A), GeoClaw uses resolutions between 1° and 20′. For the tsunami evolution from the deep sea to the continental shelf in the region around Los Angeles (Supplementary Fig. S2B) grid resolutions between 20′ and 10″ are employed in the simulations; while tsunami impact dynamics in the immediate vicinity of the Ports of Los Angeles and Long Beach are computed with resolutions varying from 10″ to 1″ (Supplementary Fig. S2, Box C). ETOPO1 elevation data[78] (1′ resolution, interpolated for resolutions < 1′) was employed for Boxes A and B. The resolution of the elevation data employed for Box C is 0.3″.

For each of the tsunami simulations, we determine the maximum tsunami amplitude relative to mean sea level (MSL) at a synthetic tide gauge in the port (see TG2 in Fig. 1c). We use a synthetic tide gauge ~50 m offshore, 17 m water depth (TG2; Fig. 1c) because smaller earthquakes and smaller tsunamis do not trigger AMR to move to the finest grid resolution at a tide gauge at the position of TG1[67], therefore not registering that a tsunami has occurred. The tide gauge location at TG2 is far enough offshore to register the lower MNTH. We omit inundation analysis from our study because of the spurious results possible in a flat port setting and the highly resolved computational grids necessary to accurately complete the analysis. Our MNTH distribution is based on the 750 individual tsunami simulations for 15 magnitudes steps between $M_w 8.0$ and 9.4 and 50 random hypocenters and earthquake areas per magnitude step. The blue histogram in Fig. 1d represents the MNTH distribution for our entire suite of earthquakes ($M_w 8.0$ to $M_w 9.4$) in the year 2000 relative to MSL with no RSLR or tidal variability included.

**Combining tsunamis, tides, and RSLR projections**. To incorporate tidal variability into our MNTH distribution, we create a tidal distribution from the Ports of Los Angeles and Long Beach tide-gauge record (TG1, Fig. 1c) from 01/1/1989 to 12/31/2019 (ID: 9410660, ref. [67]). The Ports of Los Angeles and Long Beach have a mixed semidiurnal tidal cycle in which two high and two low tides of different sizes appear every day. The maximum tidal range (MLLW-MHHW) observed at the ports is 1.68 m, while the difference between mean high water (MHW) and mean low water (MLW) is 1.16 m (estimated from the tidal records during 1983–2019). The greatest storm-driven sea level observed at the Ports of Los Angeles and Long Beach tide gauge is 1.55 m relative to MSL (10 January 2005), following the wettest 15-day period on record in Los Angeles[79]. We use a Gaussian kernel density estimate of the full tidal data to create 2000 random tidal subsamples. We linearly combine each of the tidal subsample values with each of the 50 maximum tsunami-amplitude values for the 15 magnitude steps between $M_w 8.0$ and 9.4. Androsov[38] found that in some instances, linearly combining tides and MNTH could significantly underestimate tsunami heights, and argue that tidal amplitudes and tidal currents need to be fully integrated into a tsunami model. However, the integration of tidal amplitudes and tidal currents into a tsunami model is complex and requires more research[38]. We acknowledge that our linear approach may underestimate MNTH at the ports but argue that the effect is likely much smaller than not considering tidal variability at all.

A Gaussian kernel density estimate is also employed to create 1000 subsamples from the RLSR projection data in each decade between 2000 and 2100 to incorporate RSLR into future MNTH. Every value in the matrix containing combined maximum tsunami amplitude and tidal values (red histogram on Fig. 1d) for the different magnitude steps is element-wise combined with the 1000 subsamples of RSLR values in each decade. We carried out a small parameter study to compare the method described above with a simulation framework that changes the position of the shoreline due to different sea-level rise values and simulates tsunamis in every decade. In the latter simulation framework, the tidal datum is kept constant at MHW and requires significantly more computational resources than our approach. Comparing MNTH (without tides) between the two frameworks revealed that there is a −7% to +15% difference in the MNTH in different decades due to sea-level rise.

**Relative earthquake rates along the Alaska-Aleutian subduction zone**. To assess the relative rates of earthquakes of various magnitudes along the Alaska-Aleutian subduction zone, we first determined a regional $b$-value (Supplementary Fig. S5). To calculate the $b$-value, earthquake catalogs were constructed by merging original catalogs from the U.S. Geological Survey (USGS)[80], the Geological Survey of Canada (GSC)[81], and Stover and Coffman[39]. For the USGS and GSC catalogs, online searches retrieved earthquakes since 1900 that meet the following characteristics: (a) magnitude ≥ 3.8, (b) 46°N ≤ latitude ≤ 73°N, and (c) +165° ≤ longitude ≤ −120°. Catalog processing generally followed the methodology described by ref. [39,40], with steps including reformatting, estimating a moment magnitude for each event, deleting duplicates, and declustering. The final catalog used for the analysis was spatially trimmed to the area of the Alaska-Aleutian megathrust and depth < 50 km. Magnitude-frequency distributions were modeled using the maximum-likelihood methodology of Weichert[82]. Incremental earthquake rates in 0.2-magnitude-unit bins were determined by counting earthquakes in the catalog, assuming completeness levels for M7.4 since 1900, M7.0 since 1950, M5.6 since 1964, and 5.0 since 1990. Fits to the binned rates yielded a $b$-value 0.75 for the declustered catalog (Supplementary Fig. S5).

We use the regional b-value to assess relative rates of earthquakes of various magnitudes along the Alaska-Aleutian subduction zone (Supplementary Table S2). We note that here, as in the main text, we do not use the relative earthquake rates in our modeling and instead use them to put the shift from higher to lower earthquake magnitudes needed to produce similar MNTH into context. A $b$-value of 0.75 gives an incremental rate of M8.0 earthquakes that is 6.7 times greater than the rate of M9.1 earthquakes (Supplementary Table S2). To explore the possible uncertainty in the relative frequency rate introduced by variations in the $b$-value (following Weichert[82]), we also considered $b$-values of 0.7 and 0.8 in our calculations. Using $b$-values of 0.7 and 0.8, the relative frequency of a M8.0 vs a M9.1 changes to 5.9–7.6 (i.e., 6.7 +0.9/−0.8) times, respectively. Because our consideration of $b$-value variability did not result in significant changes to the relative frequency of earthquake magnitudes, we use the declustered catalog b-value of 0.75 to calculate our relative earthquake frequencies in the main text and in Supplementary Table S2.

## Data availability

All data integral to the stated conclusions are presented within the paper and "Methods" section. Earthquake data used to calculate the relative probability of Alaska-Aleutian subduction zone earthquakes are publicly available at (http://earthquake.usgs.gov/earthquakes/search/) and (http://earthquakescanada.nrcan.gc.ca/stndon/NEDB-BNDS/bulletin-en.php).

## Code availability

The tsunami simulation code (GeoClaw) used to generate distant-source tsunami flood heights in this study is publicly available at (http://www.clawpack.org/installing.html#). The sea-level rise projections used in this study were generated using ProjectSL (https://github.com/bobkopp/ProjectSL) and LocalizeSL (https://github.com/bobkopp/LocalizeSL). The code used to combine distant-source tsunami flood heights with relative sea-level rise projections is available at (https://github.com/weiszr/PortLLALB_SLRTsun).

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

## Acknowledgements

This work was supported by funding from National Science Foundation awards to T.D. (EAR-1624795), T.D. and B.P.H. (EAR-1624533), R.W. (DGE-1735139 and GLD-1630099), A.J.G. (EAR-1625150), and R.E.K. (ICER-1663807), and from the National Aeronautics and Space Administration to R.E.K. (80NSSC17K0698). B.P.H. is also supported by the Singapore Ministry of Education Academic Research Fund MOE2019-T3-1-004 and MOE2018-T2-1-030, the National Research Foundation Singapore, and the Singapore Ministry of Education, under the Research Centers of Excellence initiative. R.C.W., R.W.B., C.S.M., and A.R.N. are supported by the Earthquake Hazard Program of the U.S. Geological Survey. This work is a contribution to PALSEA2 (Palaeo-Constraints on Sea-Level Rise) and the International Geoscience Programme (IGCP) Project 639 and 725. This work is Earth Observatory of Singapore contribution 417. Any use of trade, firm, or product names is for descriptive purposes only and does not imply endorsement by the U.S. Government.

## Author contributions

T.D., A.J.G., R.W., and B.P.H. led the writing of the main text, with contributions from other authors. T.D., R.W., and A.J.G. prepared the figures and tables. R.W. ran and analyzed the earthquake and tsunami simulations. A.J.G. and R.E.K. produced the local probabilistic sea-level rise projections. C.S.M. constructed the earthquake catalog. S.E.E., R.C.W., R.W.B, and A.R.N. contributed to the earthquake modeling and preparation of the manuscript.

## Competing interests

The authors declare no competing interests.
