## [Peer Review File · Nature Communications]

REVIEWER COMMENTS

Reviewer #1 (Remarks to the Author):

Review of paper entitled "Impacts of Alaska-Aleutian subduction zone tsunamis in California under sea-level rise", by Tina Dura and co-authors

General comments

This paper analyses the importance of subduction zone earthquake tsunamis towards California harbour subject to sea-level rise. Scenarios emerging from the Aleutian Trench are considered, and the impact towards Los Angeles port area is analysed. The authors make use of a set of multiple earthquake scenarios of various magnitudes, to show that for a site subject to relatively low hazard intensities (e.g. tsunami amplitudes $\sim 0.5-1$ m), tsunamis of these intensities will feature more frequently in the future due to climate change. This is due to the shift from large magnitude earthquakes producing such tsunami amplitudes to more moderate magnitude earthquakes. I found the paper and in particular the conclusion interesting, although the methods used are relatively standard. A set of comments to the paper are given below. Requested (major and minor) amendments and comments are discussed in separate paragraphs. Specific comments (mainly elaborating the more general first paragraphs) and a few suggested references are given in the end of the review.

The paper is highly relevant, and the conclusions are important for hazard practitioners. However, the degree of methodological novelty is limited and findings are only available for one site (Los Angeles and Long Beach). The applied methods involve quite a few simplifications (some aspects of this is discussed below), and the conclusions could be expected based on a previously published paper (Li et al, cited by the authors). The conclusion they reach seems to be relevant for a site where the tsunami amplitudes are in the order of or are lower than the sea level rise. However, a more general discussion of the possible broader impact of this analysis (e.g. other sites and sources) was lacking.

The method focuses on tsunamis induced by earthquakes from a limited segment from a single subduction zone. As sources from other segments are not considered, this may possibly lead to bias. Were similar sensitivity studies conducted for sources emerging from other segments? This is important, as the tsunami hazard would comprise a range of sources, and a limited segment would only comprise a fraction of the source frequency. In a hazard study, weighting in sources from other segments may also give different tsunami intensity ranges. Because it seems from the analysis presented that the ratio of the tsunami height to the sea-level change is important, modelling other segments might give different conclusions. This should be illuminated further.

The authors conclude that heterogeneous slip does not change the variability of the tsunami amplitudes. However, this is not in agreement with some other recent studies. The importance of random slip for far field tsunamis was first discussed by Li et al. (2016). Later, this was investigated more systematically in a broad study of historical tsunamis in the Pacific by Davis (2019). He found that it was necessary to use heterogeneous slip to comply with the variability of past earthquakes. Using a uniform slip condition was subject to bias. He further concluded that uniform slip tended to produce too small tsunamis too often, and hence provide a biased distribution. Holding together this information with the conclusion reached by the authors, it is possible that the set of sources selected may have a bias distribution compared to real events. Hence, this aspect needs more attention. As a minimum the sensitivity studies mentioned needs to be shown explicitly, and the possible bias related to the uniform slip should be discussed. The papers of Li et al. (2016) and Davies (2019) should be discussed and referenced. It is also stressed that in a previous paper (with some of the same authors as in this paper) discussing the same aspect (Li et al., 2018) heterogeneous slip was used.

An important conclusion of the paper is related to shifting from higher to lower earthquake magnitudes for future tsunami amplitudes. However, the magnitude frequency distributions these conclusions are based on are highly uncertain. But in their analysis (Figure 4 and Table 2) the authors only show deterministic curves, in cases where uncertainties are very large. It is stressed that especially for the higher magnitudes, it is

likely that conclusions largely driven by the uncertainty in the magnitude vs frequency relationships. It is therefore necessary to include such uncertainty estimation into the Figure 4 and Table 2 and discuss how these uncertainties affects conclusions, especially when moving from moderate to very large earthquake magnitudes.

The numerical simulations seem to be based on near shore points (MNTH) close to the shoreline, and omit inundation analysis. From this point, tsunamis will inundate and might amplify further. While it is fine to use such a point as a proxy for the hazard or the sensitivity, the depth and location is important to anchor the analysis. To this end, the water depth of the gauge point is a necessary input missing (at least I could not find it). How these amplitudes relate to typical tsunami run-up heights would also be interesting to note (from instance by analysing data from the historical events).

Specific comments

Lines 17-18: What about tsunamis? The focus importance of other hazards is a little distracting without mentioning tsunamis at all?

Line 36 (and generally): It is unclear what you mean by amplitude, because you don't refer to the water depth. Moreover, it would be interesting to note what would the typical related run-up height and inundation distances are?

Line 38: What are the tidal amplitudes?

Line 50: A note of caution here. 1960 has after all cause the largest heights so far. I would suggest highlighting the uncertainty aspect, and stress that other sources might contribute substantially too. But I agree that Alaska / Aleutian sources represent natural and logical examples.

Line 62 (and generally): The use of the abbreviations RSLR and MNTH makes the paper hard to read. I would suggest using more plain English terms, e.g. "sea-level rise" and "tsunami heights" or "tsunami amplitude" instead of the abbreviations, which are annoying and cumbersome to read. One could for instance introduce them early in the text.

Line 71: I understand that details needs to go in methods and supplements, but as a minimum magnitude bins and ranges should be briefly stated here. 1-2 sentences would suffice probably. Having said this, the resolution in magnitudes seems to be ok, but the homogeneous slip assumption is probably too simplified, and could represent a slight underestimation of the hazard (see above). With the unit sources of NOAA, the possibility of resolving more complex earthquakes and variable aspects ratios are limited. This must be pointed out. The sensitivity to the fault location also needs to be addressed (see above).

Line 85: At which depth is this tide gauge located? This has bearings on the conclusions.

Line 87: This does not obey the findings from recent more systematic studies (e.g. Davies 2019) which are calibrated with real events. This may point to biases in the selection of scenarios in the analysis, and this issue must be discussed and analysed further (see above).

Line 90: See above comments. The issue might be the source selection criteria and lack of heterogeneous slip.

Line 107: Was this done by adding tides to the simulations (non-linear coupling) or simply adding different heights after the simulations?

Line 128: could you please elaborate on "broadly consistent"? I.e. to which degree are they expected to deviate from IPCC projections?

Line 144: Median over what? Over all magnitudes?

Line 168: This is presented too simplistic, and uncertainty ranges needs to be added. Probability of high magnitude earthquakes are highly uncertain, and this uncertainty should be discussed here.

Figure 4: Please add uncertainties related to possible different magnitude frequency relationships.

Line 180: Same comment as line 168.

Table 2: What type of method is used to provide magnitude vs frequency relationship, and what is the respective uncertainty in the eq magnitude? It is necessary to quantify and discuss uncertainties here, as they are large and have first order effects on conclusions.

References

Davies, G. (2019). Tsunami variability from uncalibrated stochastic earthquake models: tests against deep ocean observations 2006–2016. *Geophysical Journal International*, 218(3), 1939-1960.

Li, L., Switzer, A. D., Chan, C. H., Wang, Y., Weiss, R., & Qiu, Q. (2016). How heterogeneous coseismic slip affects regional probabilistic tsunami hazard assessment: A case study in the South China Sea. *Journal of Geophysical Research: Solid Earth*, 121(8), 6250-6272.

Reviewer #2 (Remarks to the Author):

The manuscript combines a probabilistic tsunami hazard assessment approach with future climate change induced sea level rise scenarios and applies this combination to an assessment of future tsunami hazard of the ports of Los Angeles and Long Beach. While the overall approach is very relevant and the methodology in general is sound. In general the methodology is well described and the results are plausible. However, a number of weaknesses should be addressed before the material can be published. I would like to highlight only major suggestions, because the article is very well written and minor corrections or typos are almost absent in my view (I found only one typo in line 211, where the second "and" is redundant).

First of all, the idea and general approach is not new and the introduction should reflect this. For example Nagai et al. (2020) conducted a similar study for tsunami risk in Tokyo Bay. While the mentioned paper does not focus so much on the probabilistic aspect, the method to include sea level rise is similar and the authors should probably embed their study in the common scientific context.

Methodologically, it is not completely clear to me, how the sea level rise is actually accounted for. To this end I would suggest to clearly define what "tidal stage" actually means. Do the authors assume different levels of a sea-at-rest situation that correspond to future projections of sea levels with tidal levels above/below that projected mean sea level? Then, a more thorough discussion on the uncertainty introduced by this simplifying assumption should be added. It is by no means clear that the near-shore bathymetry stays unchanged with a sea level rise. It is also unclear if the near-shore tidal elevation and currents are unaffected. Androsov et al. (2011) demonstrate a non-neglectable interaction between tides and tsunami waves in certain situations. While I acknowledge the difficulty to take this into account in the simulations, it should at least be discussed as a source of uncertainty and if possible quantified.

The source modeling may be a little too simplistic. Davies (2019) demonstrates a large influence of non-homogeneous slip distribution on tsunami wave heights (even in the far field). This should certainly be considered. A homogeneous slip distribution may cause a bias on the results.

In general I am missing a thorough discussion on the uncertainties involved in this assessment. The numerous sources of uncertainty (source variability, uncertainty in sea level rise projections, influence of near-shore effects such as currents, bathymetry, etc., influence of the local grid resolution, position of the virtual tide gauge) are not discussed with enough rigour, given the stringent statements regarding results of the article.

Finally, I have some difficulty to interpret figure 4. In general, I would assume that the RCP8.5 projections indicate a larger RSLR than those related to the RCP2.6 scenarios. Additionally, DP16 appears to induce less dramatic RSLR than K14. But why do RCP2.4-K14 and RCP8.5-K14 react almost identical in MNTH=0.5m assessments, whereas the differences are dramatic in the other two significant wave height assessments? Also, why are the differences between K14 and DP16 in the MNTH=0.5m so significant, whereas in the MNTH \geq 1.0m at least for the RCP2.6 scenarios there seems to be no significant difference? I think more explanation and interpretation would be necessary.

I suggest to accept the paper after major revisions.

Jörn Behrens, Hamburg, December 29, 2020

References:

R. Nagai, T. Takabatake, M. Esteban, H. Ishii, and T. Shibayama (2020): Tsunami risk hazard in Tokyo Bay: The challenge of future sea level rise, *Int. Jou. Disaster Risk Reduction*, 45:101321, DOI:10.1016/j.ijdr.2019.101321.

A. Androsov, J. Behrens, and S. Danilov (2011): Tsunami Modelling with Unstructured Grids. Interaction between Tides and Tsunami Waves, in *Computational Science and High Performance Computing IV* (Eds. E. Krause, Y. Shokin, M. Resch, D. Kröner, and N. Shokina), Springer Berlin, Heidelberg, 115:191-206. DOI:10.1007/978-3-642-17770-5_15.

G. Davies (2019): Tsunami variability from uncalibrated stochastic earthquake models: tests against deep ocean observations 2006–2016, *Geophysical Journal International*, 218(3):1939-1960, DOI:10.1093/gji/ggz260.

Reviewer #3 (Remarks to the Author):

Title: Impacts of Alaska-Aleutian subduction zone tsunamis in California under sea-level rise

Authors: Tina Dura et al.

Review by Debi Kilb, OK to give the authors my name.

The authors investigate the impact of tsunamis generated by distant sources given the expected relative sea-level rise (RSLR). Their study focuses on large subduction zone earthquake in the Alaska-Aleutian subduction zone and the generation of a tsunami hazard in southern California. They use time-independent, deterministic earthquake modeling of Mw 8.0 – 9.4 earthquakes to explore the maximum nearshore tsunami heights (MNTH) along the shorelines of Los Angeles and Long Beach over the next century. They use a randomization approach with one caveat – they use the same 15 earthquake magnitudes and slip variations every year. I find this reasonable as it allows the authors to test the temporally changing impacts of the same set of earthquakes. The study finds that today an Mw9.1 is required to produce >1-meter MNTH in southern California, but in the future, when accounting for RSLR, the magnitude estimate is greatly reduced to Mw8. What is alarming about this finding is that Mw8.0 earthquakes are ~6 times more frequent along the Alaska-Aleutian subduction zone, which substantially elevates the seismic hazards. I expect this topic is going to be of interest to a wide range of readers.

1. A prime component of this work is the B-value used to estimate the frequencies of, for example a Mw8.0 earthquake in comparison with a Mw9.1 earthquake. The manuscript could be strengthened by including how the B-value is computed, some associated references and the addition of B-value uncertainties. With this information they can propagate these uncertainties to also include uncertainties in their final results. For example, their concluding statement “Mw8.0 earthquakes are ~6 times more frequent along the Alaska-Aleutian subduction zone than Mw9.1 earthquakes”, could be updated to include an uncertainty on the “~6 times more frequent value”. There were a number of AGU talks and posters on B-values, which I realize the authors did not have access to prior to their submission, but now they will be a good resource to strengthen the B-value component of the manuscript.
2. As I understand it there are mismatches in the numbers on lines 36-44 (explained in more detail below). Similarly, on Line 151, as I understand it, the median value listed exceeds the maximum value listed in Table 1 (explained in more detail below). Please update these numbers or include additional text to better inform the reader.

Below, I include additional minor suggestions.

Sincerely,

Debi Kilb

Detailed comments on the manuscript

-- I found the title confusing, consider revamping. One idea is below.

OLD: Impacts of Alaska-Aleutian subduction zone tsunamis in California under sea-level rise

NEW: Sea-Level Rise Impacts in California from Alaska-Aleutian subduction zone tsunamis

Line 12: OLD: RCP8.5

NEW: (called RCP8.5)

Line 16: Define <10 m coastal zone?

Line 41: Should 0.27 be 0.72 to agree with the number in Line 36? Ditto, 0.41 be changed to 0.56 to agree with Line 37? Or is there something I'm not understanding as to why the mismatch? Update for clarity.

Line 44: Again, should 0.13 on line 44 be changed to 0.31 as listed on Line 38?

Line 48: Should 'California coast' be changed to 'southern California coast'?

Line 58: Consider rewording for clarity:

OLD: in southern California and Pacific-wide tsunami hazard assessments

NEW: for Pacific-wide tsunami hazard assessments for southern California

Line 60-62: Again, for clarity consider rewording:

OLD: In order to fully understand the hazard associated with low-probability, high-impact distant-source tsunami events at the ports of Los Angeles and Long Beach, the effect of RSLR on MNTH must be considered.

NEW: The effect of RSLR on MNTH must be considered in order to fully understand the hazard associated with low-probability, high-impact distant-source tsunami events at the ports of Los Angeles and Long Beach.

Line 97: 2.5th percentile?

Line 151: The manuscript states "The medians of the MNTH distributions in 2100 reflect the wide range in emissions pathways and sea level projections, ranging from 0.95 m for RCP2.6 DP16" However in the Table 1. the range of values for 2100 for RCP2.6 DP16 range between 0.03-0.73. How, then, is it possible for the median value to exceed the maximum value listed in Table 1?

Figure 3. Add units to the x-label.

Lines 156-160: I find it very odd to include the same exact text in the main manuscript as in the caption of Figure 4. Consider removing the details from here, and instead refer to the Figure, and include this text only in the Figure caption.

SUPPLEMENTAL

- First sentence in the "Earthquake realizations" section, consider adding the magnitude step size within the bracketed information.
- Please add a sentence about how you decluster and add a reference.

- First sentence of the section “Earthquake recurrence along the Alaska-Aleutian subduction zone”. Double check is ‘ref’ the correct wording? Ditto bottom of page 4. Also, if you want to use USGS and GSC acronyms in the next sentence please include those acronyms in parenthesis in the first sentence.
- Please include how you compute b-values and add a reference. Could the uncertainties in B-values change your results substantially?
- Supplemental Figure S2: You might consider adding a where the ports of Long Beach and Los Angeles locate on Figure S2C.
- Supplemental Figure S3. Consider adding how many lines are in the figure, either within the figure (N=XXX) or in the Figure caption.

Reviewer #1 (Remarks to the Author):

Review of paper entitled "Impacts of Alaska-Aleutian subduction zone tsunamis in California under sea-level rise", by Tina Dura and co-authors

General comments: This paper analyses the importance of subduction zone earthquake tsunamis towards California harbour subject to sea-level rise. Scenarios emerging from the Aleutian Trench are considered, and the impact towards Los Angeles port area is analysed. The authors make use of a set of multiple earthquake scenarios of various magnitudes, to show that for a site subject to relatively low hazard intensities (e.g. tsunami amplitudes ~0.5-1 m), tsunamis of these intensities will feature more frequently in the future due to climate change. This is due to the shift from large magnitude earthquakes producing such tsunami amplitudes to more moderate magnitude earthquakes. I found the paper and in particular the conclusion interesting, although the methods used are relatively standard. A set of comments to the paper are given below. Requested (major and minor) amendments and comments are discussed in separate paragraphs. Specific comments (mainly elaborating the more general first paragraphs) and a few suggested references are given in the end of the review.

Reply: We thank reviewer 1 for their constructive and helpful feedback on the manuscript.

Comment: The paper is highly relevant, and the conclusions are important for hazard practitioners. However, the degree of methodological novelty is limited and findings are only available for one site (Los Angeles and Long Beach). The applied methods involve quite a few simplifications (some aspects of this is discussed below), and the conclusions could be expected based on a previously published paper (Li et al, cited by the authors).

Reply: We decided to focus on the Ports of Los Angeles and Long Beach in southern California due to 1) the economic importance of the site, and 2) the existing tsunami modeling and economic impact estimates available from the USGS SAFRR scenario. The study that the reviewer mentions by Li et al. also focused on just one site in Macau, China. The computational resources necessary for analyzing multiple sites are high, and we decided to use one site to demonstrate the changing impacts of distant-source tsunamis under future sea-level rise. In the process of our modeling efforts, we have also developed the framework for future probabilistic studies that can consider distant-source tsunamis from around the Pacific Rim.

As for the Li et al. (2018) study, we had cited it in the manuscript but we have expanded our discussion of it to address reviewer 1's concerns about uniform vs. nonuniform slip modeling on text (lines 96-98, lines 110-113) and supplementary methods (lines 121-135).

Comment: The conclusion they reach seems to be relevant for a site where the tsunami amplitudes are on the order of or are lower than the sea level rise. However, a more general discussion of the possible broader impact of this analysis (e.g. other sites and sources) was lacking.

Reply: We have added a general sentence highlighting the importance of our study on coastlines where, as the reviewer points out, the "tsunami amplitudes are on the order of or are lower than the sea-level rise." See lines 32-35 (main text).

Comment: The method focuses on tsunamis induced by earthquakes from a limited segment

from a single subduction zone. As sources from other segments are not considered, this may possibly lead to bias. Were similar sensitivity studies conducted for sources emerging from other segments? This is important, as the tsunami hazard would comprise a range of sources, and a limited segment would only comprise a fraction of the source frequency. In a hazard study, weighting in sources from other segments may also give different tsunami intensity ranges. Because it seems from the analysis presented that the ratio of the tsunami height to the sea-level change is important, modelling other segments might give different conclusions. This should be illuminated further.

Reply: We made the decision to employ a deterministic approach that focuses on segments of the Alaska-Aleutian subduction zone that, through previous modeling studies, have been shown to produce the highest “worst-case scenario” tsunami amplitudes in southern California. We purposefully did not engage in a probabilistic hazards study that would need to consider sources from around the entire Pacific Rim. We offer the following explanation on:

Lines 69-74 (main text)

“Of the distant earthquake source regions posing a tsunami threat to the California coast, the Alaska-Aleutian subduction zone has the potential to produce the highest (1-2 m) tsunami amplitudes^{16,18}. The Semidi segment of the Alaska-Aleutian subduction zone, defined as the portion of the subduction zone between western Kodiak Island and the Shumagin Islands, has been identified as a source area of particular concern because the continental slope azimuth there directs tsunamis towards southern California (Fig. 1a).”

Lines 98-103 (main text)

“Our approach differs from probabilistic tsunami hazards assessments in that we use the same source region and same suite of 15 earthquake magnitudes and slip variations every year from 2000 to 2100, without considering the probability of each earthquake magnitude. This approach allows us to consider the changing impact of the same suite of significant earthquakes and tsunamis during 21st century RSLR.”

In summary, we are using the same set of earthquakes and resultant distant-source tsunamis for every year out to 2100 to study the changing impacts of tsunamis from this particular earthquake source (Semidi and Kodiak segments) under sea-level rise.

As reviewer 3 put it, “[the authors] use a randomization approach with one caveat – they use the same 15 earthquake magnitudes and slip variations every year. I find this reasonable as it allows the authors to test the temporally changing impacts of the same set of earthquakes.”

For the purposes of highlighting distant-source tsunami impacts at the Ports of Los Angeles and Long Beach under future sea-level rise, we believe the deterministic approach we have taken in this manuscript is the most appropriate. However, in the process of our modeling efforts, we have also developed the framework for future probabilistic studies that can consider distant-source tsunamis from around the Pacific Rim.

The comments below all refer to the same question/concern from Reviewer 1, so we have grouped them below:

Comment: The authors conclude that heterogeneous slip does not change the variability of the tsunami amplitudes. However, this is not in agreement with some other recent studies. The important of random slip for far field tsunamis was first discussed by Li et al. (2016). Later, this was investigated more systematically in a broad study of historical tsunamis in the pacific by Davis (2019). He found that it was necessary to use heterogeneous slip to comply with the variability of past earthquakes. Using a uniform slip condition was subject to bias. He further concluded that uniform slip tended to produce too small tsunamis too often, and hence provide a biased distribution. Holding together this information with the conclusion reached by the authors, it is possible that the set of sources selected may have a bias distribution compared to real events. Hence, this aspect needs more attention. As a minimum the sensitivity studies mentioned needs to be shown explicitly, and the possible bias related to the uniform slip should be discussed. The papers of Li et al. (2016) and Davies (2019) should be discussed and referenced. It is also stressed that in a previous paper (with some of the same authors as in this paper) discussing the same aspect (Li et al., 2018) heterogeneous slip was used.

Comment: Line 87: This does not obey the findings from recent more systematic studies (e.g. Davies 2019) which are calibrated with real events. This may point to biases in the selection of scenarios in the analysis, and this issue must be discussed and analysed further (see above).

Comment: Line 90: See above comments. The issue might be the source selection criteria and lack of heterogeneous slip.

Comment: Having said this, the resolution in magnitudes seems to be ok, but the homogeneous slip assumption is probably too simplified, and could represent a slight underestimation of the hazard (see above). With the unit sources of NOAA, the possibility of resolving more complex earthquakes and variable aspects ratios are limited. This must be pointed out. The sensitivity to the fault location also needs to be addressed (see above).

Reply: We acknowledge that our explanation of our modeling approach was not clear enough, and we have added additional information to the main text (lines 96-98, lines 110-113) and supplementary methods (lines 121-135) to clarify (main points pasted in below). Our approach is novel in that for each magnitude step, we vary the number of unit sources over which slip can be distributed in order to simulate a variety of possible tsunami scenarios. Our approach produced a range of possible tsunamis at the ports very similar to those produced by a heterogenous slip approach, but with a fraction of the computing power. We have added Fig. S3 to show the similarity in the tsunamis at the ports using our approach and a heterogenous approach. Also, we have shown that using uniform slip across the same number of unit sources per magnitude step does underestimate tsunamis at the ports, as the reviewer mentioned it might. But, again, we do not use a traditional uniform slip approach. Please see text below and in the supplement for the explanation of our method:

Lines 96-98 (main text)

“Slip is uniform across unit sources for each earthquake; however, the area (i.e., number of unit sources) over which the slip is distributed for each earthquake magnitude can vary (supplementary methods).”

Lines 110-113 (main text)

“A small parameter study showed that our method of varying the number of unit sources over which slip is distributed for each earthquake magnitude produces a robust estimate of the MNTH distribution at the ports, similar to that of a nonuniform slip modeling approach (Fig. S3; e.g., refs^{1,15,32}).”

Lines 121-135 (supplementary methods)

“We performed a parameter study to see how the MNTH produced by our method of varying the number of unit sources over which slip is distributed for each earthquake magnitude compared to a uniform and nonuniform slip approach for the same earthquake magnitudes. Fig. S3 compares the MNTH distribution generated by earthquakes with uniform slip distributions, nonuniform slip distributions generated with the method of ref⁹, and our method described above. The comparison supports the conclusion that uniform slip (using a constant number of unit sources and slip for each magnitude while only varying the earthquake location) results in an underestimation of MNTH shown by refs^{18,20,21}. In contrast, we show that our method produces similar MNTH results to an approach that employs nonuniform slip (Fig. S3)¹⁹. We also performed a parameter study to evaluate how many earthquake areas per magnitude were necessary to produce a consistent and reproducible distribution of MNTH at the ports. Simulating a consistent distribution of MNTH is critical to ensuring that source-variability is not the driving factor in changing MNTH. In our convergence tests, we show that 50 random earthquake areas and hypocenter locations per magnitude step are sufficient to produce a robust statistical representation (i.e., consistent and reproducible) distribution of MNTH at the ports (Fig. S3).”

Also, as the reviewer requested, we have referred to the methods of Li et al. (2018) and Davies (2019) in the main text and supplement and discuss how our findings compare to theirs (main text lines 110-113 and supplementary methods lines 121-135).

Comment: Line 71: I understand that details needs to go in methods and supplements, but as a minimum magnitude bins and ranges should be briefly stated here. 1-2 sentences would suffice probably.

Reply: Moved the information about using “15 magnitude steps” to earlier in the earthquake modeling section (line 91, main text)

The comments below all refer to the same question/concern from Reviewer 1, so we have grouped them below:

Comment: An important conclusion of the paper is related to shifting from higher to lower earthquake magnitudes for future tsunami amplitudes. However, the magnitude frequency distributions these conclusions are based on are highly uncertain. But in their analysis (Figure 4 and Table 2) the authors only show deterministic curves, in cases where uncertainties are very

large. It is stressed that especially for the higher magnitudes, it is likely that conclusions largely driven by the uncertainty in the magnitude vs frequency relationships. It is therefore necessary to include such uncertainty estimation into the Figure 4 and Table 2 and discuss how these uncertainties affects conclusions, especially when moving from moderate to very large earthquake magnitudes.

Comment: Line 168: This is presented too simplistic, and uncertainty ranges needs to be added. Probability of high magnitude earthquakes are highly uncertain, and this uncertainty should be discussed here.

Comment: Figure 4: Please add uncertainties related to possible different magnitude frequency relationships.

Comment: Line 180: Same comment as line 168.

Comment: Table 2: What type of method is used to provide magnitude vs frequency relationship, and what is the respective uncertainty in the eq magnitude? It is necessary to quantify and discuss uncertainties here, as they are large and have first order effects on conclusions.

Reply: We believe there is a misunderstanding regarding our deterministic approach. There is no frequency of earthquakes considered in Figure 4 or Table 2. Each magnitude earthquake has a PDF of possible tsunami amplitudes (e.g., Fig S1) and in Figure 4 and Table 2 we are just reporting the magnitude that has a 50/50 chance of exceeding MNTH of a defined amplitude. There is no probability attached to each magnitude.

As reviewer 3 put it, “[the authors] use a randomization approach with one caveat – they use the same 15 earthquake magnitudes and slip variations every year. I find this reasonable as it allows the authors to test the temporally changing impacts of the same set of earthquakes.”

We offer the following explanation on lines 98-103 (main text):

“Our approach differs from probabilistic tsunami hazards assessments in that we use the same source region and same suite of 15 earthquake magnitudes and slip variations every year from 2000 to 2100, without considering the probability of each earthquake magnitude. This approach allows us to consider the changing impact of the same suite of significant earthquakes and tsunamis during 21st century RSLR.”

The confusion may lie in that we did analyze the earthquake catalog for the Alaska-Aleutian subduction zone to calculate the relative rates of earthquake magnitudes along the Alaska-Aleutian subduction zone. We did this to put the shift from higher to lower earthquake magnitudes into context and did not include these values in our modeling or calculations. We have added additional text to clarify our approach, pasted in below.

Lines 131-135 (main text)

“Although not included in our modeling or calculations, we analyzed the earthquake catalog along the entire Alaska-Aleutian subduction zone (ref^{87,38}; supplementary methods) in order to put the frequency of our chosen earthquake magnitudes into context. We used the declustered

earthquake catalog (b-value= 0.75) to calculate the incremental annual rate of earthquakes along the Alaska-Aleutian subduction zone (Table S2). ”

Our methods for analyzing the Alaska-Aleutian subduction zone earthquake catalog and our calculation of the b-value and uncertainties in the b-value for the margin are described in the supplementary methods (see relevant text below). Considering this uncertainty does not substantially change our results.

Lines 182-193 (supplementary methods)

“We use the regional b-value to assess relative rates of earthquakes of various magnitudes along the Alaska-Aleutian subduction zone (Table S2). We note here, as in the main text, that we do not use the relative earthquake rates in our modeling, and instead use them to put the shift from higher to lower earthquake magnitudes needed to produce similar MNTH into context. A b-value of 0.75 gives an incremental rate of M8.0 earthquakes that is 6.7 times greater than the rate of M9.1 earthquakes (Table S2). To explore the possible uncertainty in the relative frequency rate introduced by variations in the b-value (following ref^{s7}), we also considered b-values of 0.7 and 0.8 in our calculations. Using b-values of 0.7 and 0.8, the relative frequency of a M8.0 vs a M9.1 changes to 5.9–7.6 (i.e., 6.7 +0.9/-0.8) times, respectively. Because our consideration of b-value variability did not result in significant changes to the relative frequency of earthquake magnitudes, we use the declustered catalog b-value of 0.75 to calculate our relative earthquake frequencies in the main text and in Table S2.”

Comment: The numerical simulations seem to be based on near shore points (MNTH) close to the shoreline, and omit inundation analysis. From this point, tsunamis will inundate and might amplify further. While it is fine to use such a point as a proxy for the hazard or the sensitivity, the depth and location is important to anchor the analysis. To this end, the water depth of the gauge point is a necessary input missing (at least I could not find it).

Reply: As the reviewer points out, we use tide gauge two (TG2) as a proxy point to show the changing maximum nearshore tsunami heights (MNTH) at the ports. The information for the location/depth of TG2 was in the supplement, but we have now added that information to the main text for clarity (line 109). TG2 is located in the outer harbor at 17 m water depth (Fig. 1c). We have also expanded on why we chose TG2 as a reference point and why we omit inundation analysis in the main text (lines 115-123) and supplementary methods (109-120).

Comment: How these amplitudes relate to typical tsunami run-up heights would also be interesting to note (from instance by analysing data from the historical events).

Reply: We have added additional text to the main text (lines 115-123) and supplementary methods (lines 122-130) explaining why we are not doing inundation modeling in this study. Historical events had low MNTH (<0.5 m) and did not cause significant inundation of the ports but did produce damaging currents.

Specific comments

Comment: Lines 17-18: What about tsunamis? The focus importance of other hazards is a little distracting without mentioning tsunamis at all?

Reply: We open the paragraph with the most commonly considered compound hazard of storm surges and sea-level rise and finish the paragraph by stating that tsunami impacts under future sea-level rise must also be considered (lines 17-32, main text). We have added an additional sentence to the end of the opening paragraph emphasizing the importance of considering distant-source tsunami hazards under sea-level rise (lines 32-35, main text).

Comment: Line 36 (and generally): It is unclear what you mean by amplitude, because you don't refer to the water depth. Moreover, it would be interesting to note what would the typical related run-up height and inundation distances are?

Reply: We have added the definition of tsunami amplitude on lines 43-44 (main text) and specified that we are referring to historical tsunami amplitudes measured at tide gauge one in the subsequent sentences.

We have added additional information to the main text (lines 122-130) and supplement (lines 109-116) explaining why we are not doing inundation modeling in this study.

Historical events had low MNTH (<0.5 m) and did not cause significant inundation of the ports but did produce damaging currents.

Comment: Line 38: What are the tidal amplitudes?

Reply: The information about the tidal amplitude at the ports is included in Fig. 1c and in the supplementary material, but we have now included it in the main text for clarity (line 149-150). The maximum tidal range (MLLW-MHHW) observed at the Ports is 1.68 m.

Comment: Line 50: A note of caution here. 1960 has after all cause the largest heights so far. I would suggest highlighting the uncertainty aspect, and stress that other sources might contribute substantially too. But I agree that Alaska / Aleutian sources represent natural and logical examples.

Reply: In lines 39-66 we highlight other tsunami source regions (e.g., Chile, Japan) that have affected the ports and cite references (e.g., Uslu et al., 2010, Ross et al., 2013) that look at still other potential sources around the Pacific Rim (e.g., Kamchatka, Vanuatu, New Zealand). However, previous studies note that of these potential source regions, the Alaska-Aleutian subduction zone consistently returns the largest tsunami amplitudes at the ports (e.g., Uslu et al., 2010). We write in the text (line 45-46) that the Chile 1960 earthquake did indeed cause the highest tsunami amplitudes at the ports (not considering the tidal stage that it struck at, which dampened MNTH). However, we are focusing on the source region that has the potential to produce a tsunami larger than any historically observed tsunami at the ports, the Semidi and Kodiak segments of the Alaska-Aleutian subduction zone (lines 69-80, main text; Uslu et al., 2010, Ross et al., 2013). We agree with the reviewer that the Alaska-Aleutian sources represent natural and logical examples for highlighting the worst-case scenario distant-source tsunami impacts at the ports.

Comment: Line 62 (and generally): The use of the abbreviations RSLR and MNTH makes the paper hard to read. I would suggest using more plain English terms, e.g. "sea-level rise" and "tsunami heights" or "tsunami amplitude" instead of the abbreviations, which are annoying and cumbersome to read. One could for instance introduce them early in the text.

Reply: We made an effort to keep abbreviations to a minimum for readability purposes. That said, we believe the abbreviations the reviewer points out are necessary, and we would like to

preserve them in the manuscript. The abbreviation for relative sea-level rise (RSLR) is commonly used and accepted in sea level, storm, and tsunami studies. As far as the abbreviation we employ for maximum nearshore tsunami heights (MNTH), we settled on this terminology after our first round of reviews from Nature Climate Change. We had originally used “tsunami heights”, as the reviewer recommends, but that was flagged as an issue because it didn’t specify whether the tsunami heights were on land or offshore. To be more precise with what we actually mean when we report tsunamis at the ports, we settled on maximum nearshore tsunami heights (MNTH). The maximum term refers to us using the maximum amplitude relative to mean sea level when we report our tsunami heights. The nearshore term clarifies that we are using a nearshore tide gauge as a proxy point to look at the range of tsunamis that impact the ports.

Comment: Line 85: At which depth is this tide gauge located? This has bearings on the conclusions.

Reply: The information for the location/depth of TG2 was in the supplement, but we have now added that information to the main text as well for clarity (line 109). TG2 is located in the outer harbor at 17 m water depth.

Comment: Line 107: Was this done by adding tides to the simulations (non-linear coupling) or simply adding different heights after the simulations?

Reply: The latter, as outlined in the last paragraph of the supplementary methods section “Combining tsunamis, tides and RSLR projections” (lines 138-154).

Comment: Line 128: could you please elaborate on "broadly consistent"? I.e. to which degree are they expected to deviate from IPCC projections?

Reply: We have removed the word “broadly” to clarify that they are consistent with projections. Taken from the text of Kopp et al. (2014) “. . . our *likely* projections of global sea level in 2100 are close to those of AR5 . . . , though differ slightly (e.g., in RCP8.5 in 2100, 0.6-1.0m vs. AR5’s 0.5-1.0m)”

They note that differences are likely due to their handling of thermal expansion (K14 includes a drift correction to a possibly non-zero background thermal expansion), the use of Marzeion et al. (2012) for glaciers and ice cap contribution in K14, and differing baselines (K14 use the year 2000 as a baseline, where as AR5 used 1985-2005).

Comment: Line 144: Median over what? Over all magnitudes?

Reply: Yes, over all magnitudes. We have added “from our full suite of earthquakes to line 208 (main text) for clarity.

Comment: References

Davies, G. (2019). Tsunami variability from uncalibrated stochastic earthquake models: tests against deep ocean observations 2006–2016. *Geophysical Journal International*, 218(3), 1939-1960.

Li, L., Switzer, A. D., Chan, C. H., Wang, Y., Weiss, R., & Qiu, Q. (2016). How heterogeneous coseismic slip affects regional probabilistic tsunami hazard assessment: A case study in the

South China Sea. Journal of Geophysical Research: Solid Earth, 121(8), 6250-6272.

Reply: We have added the above references to the manuscript.

Reviewer #2 (Remarks to the Author):

Comment: The manuscript combines a probabilistic tsunami hazard assessment approach with future climate change induced sea level rise scenarios and applies this combination to an assessment of future tsunami hazard of the ports of Los Angeles and Long Beach. While the overall approach is very relevant and the methodology in general is sound. In general the methodology is well described and the results are plausible. However, a number of weaknesses should be addressed before the material can be published. I would like to highlight only major suggestions, because the article is very well written and minor corrections or typos are almost absent in my view (I found only one typo in line 211, where the second "and" is redundant).

First of all, the idea and general approach is not new and the introduction should reflect this. For example Nagai et al. (2020) conducted a similar study for tsunami risk in Tokyo Bay. While the mentioned paper does not focus so much on the probabilistic aspect, the method to include sea level rise is similar and the authors should probably embed their study in the common scientific context.

Reply: We have added the Nagai et al. (2020) reference to the introduction (line 32). However, we note that our method of incorporating a subsample of low and high-end values of sea-level rise projections (Fig. 2, lines 198-207 main text and lines 155-165 supplementary methods) is different than the method of simply raising the baseline sea-level by the mean of sea level projections like was done in Nagai et al., (2020). Our subsampling of the two sets of RSLR projections taken together provides a reasonable approach to bracketing the range of plausible probability distributions of future rise.

Comment: Methodologically, it is not completely clear to me, how the sea level rise is actually accounted for. To this end I would suggest to clearly define what "tidal stage" actually means. Do the authors assume different levels of a sea-at-rest situation that correspond to future projections of sea levels with tidal levels above/below that projected mean sea level?

Reply: Yes, the reviewer is correct in their summary of our approach. Please see main text lines 145-146 and 198-207, and supplementary methods lines 138-165.

Comment: Then, a more thorough discussion on the uncertainty introduced by this simplifying assumption should be added. It is by no means clear that the near-shore bathymetry stays unchanged with a sea level rise.

Reply: We have addressed this comment by adding text about the total uncertainty introduced by the linear combination of sea-level rise projections and tidal variability with MNTS to the main text (lines 198-207) and elaborated in the supplementary methods (lines 138-165).

Comment: It is also unclear if the near-shore tidal elevation and currents are unaffected. Androsov et al. (2011) demonstrate a non-neglectable interaction between tides and tsunami waves in certain situations. While I acknowledge the difficulty to take this into account in the

simulations, it should at least be discussed as a source of uncertainty and if possible quantified.

Reply: The reviewer is correct that the interaction between tides and tsunami waves can affect MNTH. We have addressed how tidal variations influence MNTH in a simple way, by incorporating tidal variability into our MNTH modeling, but as the reviewer points out, exploring the interaction of tidal currents with an arriving tsunami is complex and difficult to incorporate into our simulations. We have added a few lines to the main text explaining why we don't undertake this complex modeling (lines 122-130, main text) and supplementary methods (lines 147-154).

We consulted the Androsov et al. (2011) reference the reviewer pointed out, and from what we could gather, the modeling in the paper found that tidal currents in some situations can amplify tsunami waves. Thus, our approach in this paper may be slightly underestimating possible MNTH. However, the Androsov et al. (2011) paper emphasized that more modeling and research needs to be done to understand the interaction of tidal currents and tsunami waves, thus it is difficult for us to quantify the effect of the non-linear tidal interaction. Instead, we have added a sentence to the main text stating that our MNTH may be underestimated due to not considering this effect (lines 129-130).

Comment: The source modeling may be a little too simplistic. Davies (2019) demonstrates a large influence of non-homogeneous slip distribution on tsunami wave heights (even in the far field). This should certainly be considered. A homogeneous slip distribution may cause a bias on the results.

Reply: Please see response to Reviewer 1 on this same topic.

See main text (lines 96-98, lines 110-113), supplementary methods (lines 121-135), and new Fig. S3.

Comment: In general I am missing a thorough discussion on the uncertainties involved in this assessment. The numerous sources of uncertainty (source variability, uncertainty in sea level rise projections, influence of near-shore effects such as currents, bathymetry, etc., influence of the local grid resolution, position of the virtual tide gauge) are not discussed with enough rigour, given the stringent statements regarding results of the article.

Reply: We have added additional information to the main text and supplement discussing our uncertainties. See below for the line numbers corresponding to uncertainty discussion:

Source variability: In lines 110-121 (main text), lines 121-135 (supplementary methods), and Fig. S4 we explain how we considered the possibility of source variability producing uncertainty in our MNTH. In our parameter study we showed that 50 earthquakes per magnitude step generates a reproducible distribution of MNTH at the ports.

Sea-level rise projections: We consider four future RSLR probability distributions: two each for RCP2.6 and RCP 8.5 emission scenarios, where each scenario includes one set of SLR projections with ice sheet contributions in line with projections from AR5, and one set of SLR projections that accounts for potentially accelerated collapse of portions of the Antarctic Ice Sheet. Taken together, this group of RSLR projections provide a reasonable approach to bracketing the range of plausible probability distributions of future rise (see Fig. 2 for the 95% confidence interval of each projection, lines 158-184 main text, 2-41 supplementary methods).

The sea level projections consider processes such as atmosphere-ocean dynamics, the gravitational, rotational, and dynamic effects of ocean/cryosphere/hydrosphere mass redistribution, glacio-isostatic adjustment, sediment compaction, and tectonic uplift or subsidence to generate local probability distributions of RSLR (Kopp et al., 2014; Kopp et al., 2017). Our approach of using a distribution of possible future sea levels (198-207, main text) in our MNTH distributions differs from previous studies incorporating sea-level rise into tsunami heights (e.g., Li et al., 2018; Nagai et al. 2020) that used the mean of RSLR projections to raise sea-level in their modeling. We believe our approach better accounts for uncertainty in future RSLR.

Influence of nearshore effects such as current, bathymetry etc: In lines 122-129 (main text) we discuss our decision not to include complex processes such as coastal inundation, currents in the ports, or the interaction of tidal currents and tsunami waves in our modeling. If anything, not including these processes underestimates the effects of MNTH at the ports. We have added information on lines 129-130 and 202-207 (main text) discussing possible nearshore effects on MNTH.

Influence of local grid resolution: We use a commonly used (e.g., Li et al., 2018) grid resolution (lines 91-108, supplementary methods).

Position of the virtual tide gauge: We explain how the position of the virtual tide gauge ~50 m offshore at 17 m water depth helps remove uncertainty in MNTH in lines 109-116 (supplementary methods). We use tide gauge two (TG2) as a proxy point to show the changing maximum nearshore tsunami heights (MNTH) at the ports under future RSLR.

Comment: Finally, I have some difficulty to interpret figure 4. In general, I would assume that the RCP8.5 projections indicate a larger RSLR than those related to the RCP2.6 scenarios. Additionally, DP16 appears to induce less dramatic RSLR than K14. But why do RCP2.4-K14 and RCP8.5-K14 react almost identical in MNTH=0.5m assessments, whereas the differences are dramatic in the other two significant wave height assessments? Also, why are the differences between K14 and DP16 in the MNTH=0.5m so significant, whereas in the MNTH \geq 1.0m at least for the RCP2.6 scenarios there seems to be no significant difference? I think more explanation and interpretation would be necessary.

Reply: This plot is driven by RSLR and is consistent with subtle differences in RSLR projections as outlined in Kopp et al., 2017. The reason for this is that K14 projections assume a simple linear change in rate of mass loss for AIS which causes a perfect correlation between AIS contributions in the near term and that projected for the long term, but DP16 projections have no such assumption, and show no correlation between near term contributions from the AIS and long term contributions.

So, the “more complex temporal dynamics of the DP16 simulations” (Kopp et al., 2017), basically means that, through mid-century there’s actually very little difference between low SLR projections and higher SLR projections—though the projections quickly diverge later in time, with more significant contributions from the AIS.

In other words, we wouldn't necessarily expect a lot of difference between RCP8.5 DP16 and RCP2.6 DP16 until about 2040-2050 (this is consistent with our Fig. 2). So, given that the RCP2.6 K14 projections slightly outpace the RCP2.6 DP16 projections, it's not unreasonable that the RCP2.6 K14 projections might also slightly outpace the RCP8.5 DP16 projections early in this century . . . but we'd expect the two pathways to quickly diverge in the future.

Comment: References:

R. Nagai, T. Takabatake, M. Esteban, H. Ishii, and T. Shibayama (2020): Tsunami risk hazard in Tokyo Bay: The challenge of future sea level rise, *Int. Jou. Disaster Risk Reduction*, 45:101321, DOI:10.1016/j.ijdr.2019.101321.

A. Androsof, J. Behrens, and S. Danilov (2011): Tsunami Modelling with Unstructured Grids. Interaction between Tides and Tsunami Waves, in *Computational Science and High Performance Computing IV* (Eds. E. Krause, Y. Shokin, M. Resch, D. Kröner, and N. Shokina), Springer Berlin, Heidelberg, 115:191-206. DOI:10.1007/978-3-642-17770-5_15.

G. Davies (2019): Tsunami variability from uncalibrated stochastic earthquake models: tests against deep ocean observations 2006–2016, *Geophysical Journal International*, 218(3):1939-1960, DOI:10.1093/gji/ggz260.

Reply: We have included the papers suggested above in the main text.

Reviewer #3 (Remarks to the Author):

The authors investigate the impact of tsunamis generated by distant sources given the expected relative sea-level rise (RSLR). Their study focuses on large subduction zone earthquake in the Alaska-Aleutian subduction zone and the generation of a tsunami hazard in southern California. They use time-independent, deterministic earthquake modeling of Mw 8.0 – 9.4 earthquakes to explore the maximum nearshore tsunami heights (MNTH) along the shorelines of Los Angeles and Long Beach over the next century. They use a randomization approach with one caveat – they use the same 15 earthquake magnitudes and slip variations every year. I find this reasonable as it allows the authors to test the temporally changing impacts of the same set of earthquakes. The study finds that today an Mw9.1 is required to produce >1-meter MNTH in southern California, but in the future, when accounting for RSLR, the magnitude estimate is greatly reduced to Mw8. What is alarming about this finding is that Mw8.0 earthquakes are ~6 times more frequent along the Alaska-Aleutian subduction zone, which substantially elevates the seismic hazards. I expect this topic is going to be of interest to a wide range of readers.

Comment: 1. A prime component of this work is the B-value used to estimate the frequencies of, for example a Mw8.0 earthquake in comparison with a Mw9.1 earthquake. The manuscript could be strengthened by including how the B-value is computed, some associated references and the addition of B-value uncertainties. With this information they can propagate these uncertainties to also include uncertainties in their final results. For example, their concluding statement “Mw8.0 earthquakes are ~6 times more frequent along the Alaska-Aleutian subduction zone than Mw9.1 earthquakes”, could be updated to include an uncertainty on the “~6 times more frequent value”. There were a number of AGU talks and posters on B-values, which I realize the authors did not have access to

prior to their submission, but now they will be a good resource to strengthen the B-value component of the manuscript.

Reply: Our methods for analyzing the Alaska-Aleutian subduction zone earthquake catalog and our calculation of the b-value (Weichert reference) and uncertainties in the b-value for the margin are described in the supplementary methods (see relevant text below). Considering this uncertainty does not substantially change our results.

Lines 182-193 (supplementary methods)

“We use the regional b-value to assess relative rates of earthquakes of various magnitudes along the Alaska-Aleutian subduction zone (Table S2). We note here, as in the main text, that we do not use the relative earthquake rates in our modeling, and instead use them to put the shift from higher to lower earthquake magnitudes needed to produce similar MNTH into context. A b-value of 0.75 gives an incremental rate of M8.0 earthquakes that is 6.7 times greater than the rate of M9.1 earthquakes (Table S2). To explore the possible uncertainty in the relative frequency rate introduced by variations in the b-value (following ref³⁷), we also considered b-values of 0.7 and 0.8 in our calculations. Using b-values of 0.7 and 0.8, the relative frequency of a M8.0 vs a M9.1 changes to 5.9–7.6 (i.e., 6.7 +0.9/-0.8) times, respectively. Because our consideration of b-value variability did not result in significant changes to the relative frequency of earthquake magnitudes, we use the declustered catalog b-value of 0.75 to calculate our relative earthquake frequencies in the main text and in Table S2.”

Because the incremental rate of earthquakes is not used in our modeling of earthquakes, we don't propagate that uncertainty throughout our analyses. We simply used the relative rate of earthquakes to put our results into a general context. We offer the following explanation:

Lines 98-103 (main text):

“Our approach differs from probabilistic tsunami hazards assessments in that we use the same source region and same suite of 15 earthquake magnitudes and slip variations every year from 2000 to 2100, without considering the probability of each earthquake magnitude. This approach allows us to consider the changing impact of the same suite of significant earthquakes and tsunamis during 21st century RSLR.”

Lines 131-135 (main text)

“Although not included in our modeling or calculations, we analyzed the earthquake catalog along the entire Alaska-Aleutian subduction zone (ref^{37,38}; supplementary methods) in order to put the frequency of our chosen earthquake magnitudes into context. We used the declustered earthquake catalog (b-value= 0.75) to calculate the incremental annual rate of earthquakes along the Alaska-Aleutian subduction zone (Table S2).”

Comment: 2. As I understand it there are mismatches in the numbers on lines 36-44 (explained in more detail below). Similarly, on Line 151, as I understand it, the median value listed exceeds the maximum value listed in Table 1 (explained in more detail below). Please update these numbers or include additional text to better inform the reader.

Reply: Please see responses to these queries below.

Comment: Line 41: Should 0.27 be 0.72 to agree with the number in Line 36? Ditto, 0.41 be changed to 0.56 to agree with Line 37? Or is there something I'm not understanding as to why the mismatch? Update for clarity.

Reply: The first set of values are the wave amplitudes as measured at TG1 at the Ports of Los Angeles and Long beach. These values do not consider the tidal stage the tsunami arrived at. The second set of values considers the tidal stage the tsunamis arrived at (in all three cases, the tsunami hit at low tide), so the amplitudes of the tsunamis are lower than the original amplitudes (lines 48-66).

Comment: Line 44: Again, should 0.13 on line 44 be changed to 0.31 as listed on Line 38?

Reply: The value is fine as is. Please see response above for clarification.

Comment: Line 48: Should 'California coast' be changed to 'southern California coast'?

Reply: The text is correct as is. We wanted to point out that the damage estimates we were reporting are not from southern California where the tsunami didn't produce much damage, but they are instead from northern California.

Comment: Line 151: The manuscript states "The medians of the MNTH distributions in 2100 reflect the wide range in emissions pathways and sea level projections, ranging from 0.95 m for RCP2.6 DP16" However in the Table 1. the range of values for 2100 for RCP2.6 DP16 range between 0.03-0.73. How, then, is it possible for the median value to exceed the maximum value listed in Table 1?

Reply: The medians we are referring to are the medians of the MNTH distributions, not the medians of the sea-level rise projections. The values should be ok as they are.

Comment: -- I found the title confusing, consider revamping. One idea is below.

OLD: Impacts of Alaska-Aleutian subduction zone tsunamis in California under sea-level rise

NEW: Sea-Level Rise Impacts in California from Alaska-Aleutian subduction zone Tsunamis

Reply: We believe the original title better captures the changing impact of the tsunamis under future RSLR.

With a few extra characters, we could update the title to "Changing impacts of Alaska-Aleutian subduction zone tsunamis in California under future sea-level rise", which we believe would be a stronger title.

Comment: Line 12: OLD: RCP8.5

NEW: (called RCP8.5)

Reply: We have clarified what the abbreviation stands for.

Comment: Line 16: Define <10 m coastal zone?

Reply: The <10 m coastal zone has been defined.

Comment: Line 58: Consider rewording for clarity:

OLD: in southern California and Pacific-wide tsunami hazard assessments

NEW: for Pacific-wide tsunami hazard assessments for southern California

Reply: We believe the proposed edit changes the intended meaning of the sentence, so we will leave the text as is.

Comment: Line 60-62: Again, for clarity consider rewording:

OLD: In order to fully understand the hazard associated with low-probability, high-impact distant-source tsunami events at the ports of Los Angeles and Long Beach, the effect of RSLR on MNTH must be considered.

NEW: The effect of RSLR on MNTH must be considered in order to fully understand the hazard associated with low-probability, high-impact distant-source tsunami events at the ports of Los Angeles and Long Beach.

Reply: We attempted to update the topic sentence but eventually circled back around to the original sentence. Because the previous paragraphs are discussing the tsunami threat to southern California, it makes more sense to begin the topic sentence of the following paragraph with how that tsunami threat must be considered in the context of RSLR.

Comment: Line 97: 2.5th percentile?

Reply: We have changed this to 95% central range (CR) throughout.

Comment: Figure 3. Add units to the x-label.

Reply: We have added units to the x-axis

Comment: Lines 156-160: I find it very odd to include the same exact text in the main manuscript as in the caption of Figure 4. Consider removing the details from here, and instead refer to the Figure, and include this text only in the Figure caption.

Reply: We have removed the duplicate text from the figure caption and left it in the main text.

SUPPLEMENTAL

Comment: First sentence in the “Earthquake realizations” section, consider adding the magnitude step size within the bracketed information.

Reply: We have added the magnitude step size to the bracketed information (line 45 supplementary methods).

Comment: Please add a sentence about how you decluster and add a reference.

Reply: We have added additional information about our b-value calculations and consideration of uncertainty to the supplement.

Comment: First sentence of the section “Earthquake recurrence along the Alaska-Aleutian subduction zone”. Double check is ‘ref’ the correct wording? Ditto bottom of page 4. Also, if you want to use USGS and GSC acronyms in the next sentence please include those acronyms in parenthesis in the first sentence.

Reply: We have edited the text following the reviewer’s recommendations.

Comment: Please include how you compute b-values and add a reference. Could the uncertainties in B-values change your results substantially?

Reply: We described our methods for calculating b-values in the supplement (Weichert reference). We have added a paragraph that considers the uncertainty in b-values. Considering this uncertainty does not substantially change our results (lines 168-193 supplementary methods).

Comment: Supplemental Figure S2: You might consider adding a where the ports of Long Beach and Los Angeles locate on Figure S2C.

Reply: We have added more information to the caption that defines the area shown in each box in Fig. S2.

Comment: Supplemental Figure S3. Consider adding how many lines are in the figure, either within the figure (N=XXX) or in the Figure caption.

Reply: (a) Cumulative density functions of maximum tsunami elevations at the tide gauge for all magnitudes. The different lines (100 total) represent different numbers of earthquakes per magnitude step, from 1 earthquake per magnitude step to 100 earthquakes per magnitude step. In (b) we show that 50 earthquakes per magnitude step generate a reproducible MINTH distribution at the ports.

REVIEWER COMMENTS

Reviewer #1 (Remarks to the Author):

Review of revised paper entitled "Changing impacts of Alaska-Aleutian subduction zone tsunamis in California under future sea-level rise", by Tina Dura and co-authors

The new version of the paper along with the rebuttal letter clarifies a lot of issues concerning the first review. Clarifications related to details of several metrics such as tsunami amplitudes also make the analysis much more transparent, and it was easier to follow the paper's logic. Still, a few important open issues remain, concerning primarily two major issues:

1. The quantification of and reporting relative earthquake frequencies should be removed from the paper for the following reason: The analysis concerning the large magnitude uncertainties of annual occurrence rates are not properly considered. The authors report relative differences in earthquake frequency using a short (~50 year) geological record to derive G-R relationships. The probabilities of the large magnitude earthquakes they report go far beyond the validity record, and are highly uncertain. For large magnitudes, tapering becomes important and it is not sufficient to use only the b-value (see e.g. Grezio et al., 2017). Moreover, reporting relative earthquake frequencies introduces the concept of hazard (probability of occurrence) which the authors rightfully clarify that they don't aim to do. I do not think the discussion about earthquake frequency is necessary, the sensitivity to M_w on tsunami amplitudes is sufficient to make the paper relevant.

2. It seems that there is still bias related to non-uniform slip and that several statements related to this are not fully justified. To this end, I still cannot figure out completely the distinction between what you refer to as "our method" and the use of non-uniform slip. This should be crystal clear and appear in the main body of the text. From my reading of the supplement, it seems your method aligns with the concepts of taking into account variable source areas (still with uniform slip), coined "VAUS" in Davies et al. (2019). Holding together this with the findings of Davies et al. (2019), the "VAUS" treatment of sources is clearly better than uniform slip but is still introducing bias (lower average tsunami heights). This limitation should be noted within the limitations of the study (e.g. together with other modelling limitations). Fig S3 follows this pattern; the results using "our method" seem to lie in between results using uniform and non-uniform slip, hence with a lower average tsunami height. Hence, the statement in the figure caption "Our method produces similar M_{NTH} results to an approach that employs nonuniform slip" is not justified. Therefore, it is likely there is still bias related to this simplification, and related statements and conclusions throughout the paper need to be softened to take this into account. On the other hand, this also indicates that the analysis results are likely on the low (conservative) side, tsunami amplitudes could be even higher with proper inclusion of variable slip.

Line-by-line comments

Line 14: Remove statement about earthquake frequency "~6.7 times more frequent"

Line 71: It is not clear if these cited reports are the most updated reports fully addressing the hazard (probabilistically), the references are dated 10 or more years back and there has been substantial development since then. Please consider possible newer references, if available.

Line 97: Are you superposing several uniform slip unit sources, or using single unit sources with uniform slip but with different areas? The explanation is not completely clear. Lack of use of earthquake scaling laws (ala "VAUS" sources in Davies et al. 2019) is a clear limitation if uniform slip is being used.

Line 111: You do not demonstrate anywhere that this is a robust estimate. A robust estimate would recreate observed tsunami variability. Please remove this statement. Fig S3 clearly shows that the method of Li et al. gives a broader distribution. Hence, you do not fully cover this uncertainty. Rather than stating that this is a "robust estimate", it should be discussed among other limitations (such as on the next page)

Lines 132-135: I suggest removing this part. Tsunami frequency for high magnitude earthquakes are strongly determined by tapering and uncertainties, and this is not considered in this paper.

Lines 233, 247, 250, 264: Please remove statements of earthquake probability. As substantiated above, this is misleading, but also not necessary for this paper (sensitivity to M_w alone is sufficient).

References

Grezio, A., et al., (2017). Probabilistic tsunami hazard analysis: Multiple sources and global applications. *Reviews of Geophysics*, 55(4), 1158-1198.

Reviewer #3 (Remarks to the Author):

Manuscript Number: NCOMMS-20-45056-T

New Title: Changing impacts of Alaska-Aleutian subduction zone tsunamis in California under future sea-level rise

Authors: Tina Dura et al.

Review by Debi Kilb, OK to give the authors my name.

This is the second time I am reviewing this article. I was impressed with the authors detailed response to the reviews, they carefully considered the reviewers comments and very clearly outlined changes and updates and carefully described why they disagreed with a few of the reviewers' suggestions.

When I first read this manuscript, my biggest concern was if uncertainties in the B-values were propagated through to the final results that the results would not remain robust. The authors examined this by assuming ± 0.1 uncertainties in B-values and determined that tests using the wider range of B-values produced results that did not change significantly. I appreciate the authors looking into this matter and I think this update makes the paper stronger.

A secondary concern of mine on the first reading of this manuscript was my confusion as to why the median value listed on line 151 of the original manuscript exceeded the maximum value listed in Table 1 in the original manuscript. The authors explained that the median value on line 151 referred to the medians of the MNTD distributions, not the medians of the sea-level rise projection. This clears up my confusion. If they have room, I suggest this distinction be added to the manuscript, but I don't feel all that strongly about the addition.

I like the new title.

I look forward to seeing this manuscript published in Nature Communications.

Dr. Debi Kilb
Scripps Institution of Oceanography

We address the remaining concerns of Reviewer #1 below with inline replies. We have broken up the comments to help focus the discussion.

Reviewer #1 (Remarks to the Author):

Review of revised paper entitled "Changing impacts of Alaska-Aleutian subduction zone tsunamis in California under future sea-level rise", by Tina Dura and co-authors

The new version of the paper along with the rebuttal letter clarifies a lot of issues concerning the first review. Clarifications related to details of several metrics such tsunami amplitudes also make the analysis much more transparent, and it was easier to follow the papers logic.

We thank Reviewer #1 for their continued constructive and detailed feedback on the manuscript.

Still, a few important open issues remain, concerning primarily two major issues:

1. The quantification of and reporting relative earthquake frequencies should be removed from paper for the following reason:

Addressed. We have added a sentence and references (Grezio et al., 2017 and Davies et al., 2018) to the manuscript to address the reviewer's concerns about the representation of large magnitude events (lines 131-134).

However, we respectfully disagree with the essence of this comment on fundamental scientific grounds and we have not removed the discussion of relative earthquake frequencies. We believe our approach is appropriate and reasonable, and that relative earthquake rates provide important context for the earthquake magnitudes we discuss and emphasize the significance of the study. Reviewers #2 and #3 also agree that our approach is reasonable. The relative rates we report provide necessary context for our results and emphasize the significance in the drop in magnitude necessary to exceed flood height under RSLR. We respond in detail below.

The analysis concerning the large magnitude uncertainties of annual occurrence rates are not properly considered. The authors report relative differences in earthquake frequency using a short (~50 year) geological record to derive G-R relationships. The probabilities of the large magnitudes earthquakes they report goes far beyond the validity record, and are highly uncertain.

Addressed. The Reviewer is mistaken that the geological record is used to derive G-R relationships. Instead, we use among the most complete seismologic records available for a subduction zone, which includes historical events up to magnitude 9.2. The Reviewer appears to have confused absolute probabilities with relative probabilities for a given earthquake magnitude, and the relative probabilities are well-characterized by the available earthquake catalog and our Gutenberg-Richter analysis. We have added text and references to address this point (see above).

Our focus is on relative probabilities, and we have addressed uncertainties in a reasonable way (see comments by Reviewer #3, below, and new text at lines 131-134). We use a simple, but widely accepted approach well within the mainstream of earthquake statistics. We are confused by the concept of a “validity record” mentioned by the Reviewer, which is not a concept in general use in the earthquake statistics community.

For large magnitudes, tapering becomes important and it is not sufficient to use only the b-value (see e.g. Grezio et al., 2017).

No change. Unfortunately, the reviewer’s concerns are unclear to us here. If the concern is that we are overestimating the occurrence of the largest events, the reviewer may be confused about the implications of our use of the regional b-value from an existing catalog: We are using the most reasonable and defensible approach that will not overestimate large magnitudes. The *absolute* frequency of higher magnitude earthquakes can be hard to constrain anywhere due to undersampling – this is a fundamental problem in earthquake science. However, the Alaska-Aleutian subduction zone has robust seismic catalog from M5-M9.2 so using the b-value in this case to estimate of the *relative* recurrence of large-magnitude events is appropriate.

The reviewer points to the Grezio et al. (2017) review paper, which states that using the Gutenberg-Richter relationship to extrapolate the number of large earthquakes from the rate of smaller magnitude earthquakes may “strongly underpredict” large earthquakes. This is a well-known issue and is most often a problem with limited seismic catalogs for individual fault systems, but we are examining a robust regional catalog for a plate boundary system. The Grezio et al. (2017) concern about underpredicting large earthquakes is strong support for our approach of using the regional b-value, which is a more transparent and cautious way to calculate relative earthquake rates than modeling the rates of high-magnitude events (e.g. tapering or some other *a priori* approach). We are confident we are not overestimating the relative occurrence of the largest events.

Moreover, reporting relative earthquake frequencies introduces the concept of hazard (probability of occurrence) which the authors rightfully clarify that they don’t aim to do.

I do not think the discussion about earthquake frequency is necessary, the sensitivity to Mw on tsunami amplitudes is sufficient to make the paper relevant.

No change. This comment is a bit of a red herring, because discussing relative probabilities of earthquake magnitudes from a straightforward G-R analysis is clearly not moving into full-blown hazard (e.g. probability of an event or exceedance at some probability at a given location in a defined time-frame). We do not make such calculations here, which is also made clear in the paper (lines 127-129). The reviewer holds the opinion that the discussion of relative earthquake frequency is not necessary; we strongly disagree with this opinion.

We must add that we find it troubling that Reviewer #1 asked for the relative earthquake rate portion of the paper to be removed only in the second round of reviews. It's hard not to suspect that the Reviewer is moving the goalposts. We have satisfied the other two Reviewers on this point and Reviewer #1 did not note this as a major issue in the first round of reviews. We hope we have demonstrated that the issues brought up by Reviewer #1 are primarily differences in opinion and not technical problems (see above), and that by adding a slight addition to the main text, we have provided a satisfactory response to the Reviewer and the larger community on this point.

2. It seems that there is still bias related to non-uniform slip and that several statements related to this is not fully justified. To this end, I still cannot figure completely the distinction between what you refer to as "our method" and the use of non-uniform slip. This should be crystal clear and appear in the main body of the text. From my reading of supplement, it seems your method aligns with the concepts of taking into account variable source areas (still with uniform slip), coined "VAUS" in Davies et al. (2019). Holding together this with the findings of Davies et al. (2019), the "VAUS" treatment of sources is clearly better than uniform slip but is still introducing bias (lower average tsunami heights). This limitation should be noted within the limitations of the study (e.g. together with other modelling limitations).

Addressed: We have softened our language to acknowledge that the chosen modeling approach may underestimate the extreme high values of MNTH (lines 105-111). We agree with the reviewer that this is a significant improvement from using a simple uniform slip approach. We have also moved the detailed description of our parameter study from the supplement to the main text to emphasize the comparison of the different modeling approaches (lines 102-117), also pasted in below.

Lines 102-117 (main text)

"We performed a parameter study to see how the MNTH produced by our method of varying the number of unit sources over which slip is distributed for each earthquake magnitude compared to a uniform and nonuniform slip approach for the same earthquake magnitudes (Fig. S3). The parameter study supports the conclusion that uniform slip (using a constant number of unit sources and slip for each magnitude while only varying the earthquake location) results in a significant underestimation of MNTH^{34,35}. Our method produces a broader range of MNTH similar to a nonuniform slip approach, although there may be an underestimation of the higher extremes of possible MNTH^{34,35}(Fig. S3). Therefore, the MNTH described in this paper provide a conservative

estimate of possible tsunami impacts at the ports. We also performed a parameter study to evaluate how many earthquake areas per magnitude were necessary to produce a consistent and reproducible distribution of MNTH at the ports. Simulating a consistent distribution of MNTH is critical to ensuring that source-variability is not the driving factor in changing MNTH. We show that 50 random earthquake areas and hypocenter locations per magnitude step are sufficient to produce a robust statistical representation (i.e., consistent and reproducible) distribution of MNTH at the ports (Fig. S4; supplementary methods)."

Fig S3 follows this pattern; the results using "our method" seems to lie in between results using uniform and non-uniform slip, hence with a lower average tsunami height. Hence, the statement in the figures caption "Our method produces similar MNTH results to an approach that employs nonuniform slip" is not justified.

Addressed: Please see response above; we believe the new main text better supports the fact that the method produces 'similar MNTH results.' We have also adjusted the caption for Fig S3 to reflect our potential underestimation of the higher extremes of MNTH.

Therefore, it is likely some still bias related to this simplification, and related statements and conclusions throughout the paper needs to be softened to take this into account. On the other hand, this also indicates that the analysis results likely is on low (conservative) side, tsunami amplitudes could be even higher with proper inclusion of variable slip.

Addressed: We have included the following lines to acknowledge the underestimation of MNTH:

Lines 107-111

"Our method produces a broader range of MNTH similar to a nonuniform slip approach, although there may be an underestimation of the higher extremes of possible MNTH^{34,35} (Fig. S3). Therefore, the MNTH described in this paper provide a conservative estimate of possible tsunami impacts at the ports."

Line-by-line comments

Line 14: Remove statement about earthquake frequency "~6.7 times more frequent"

Addressed: We have made the choice to retain the earthquake frequency relationships. Our focus is on relative probabilities, and we have addressed uncertainties in a reasonable way (see comments by Reviewer #3, below, and new text at lines 131-134. We use a simple, but widely accepted approach well within the mainstream of earthquake statistics. We have explained our reasoning in more detail in the response to reviewer point (1) above.

Line 71: It is not clear if these cited reports are the most updated reports fully addressing the hazard (probabilistically), the references are dated 10 or more years back and there has been substantial development since then. Please consider possible newer references, if available.

Addressed: The two references we included are some of the most recent that focused on the impact of Alaska-Aleutian subduction zone ruptures on the southern California coast, specifically. To be more complete, we have added a new reference (Kalligeris et al., 2017) that also explored the topic:

Kalligeris, N., Montoya, L., Ayca, A. & Lynett, P. An approach for estimating the largest probable tsunami from far-field subduction zone earthquakes. *Nat. Hazards* **89**, 233–253 (2017).

Line 97: Are you superposing several uniform slip unit sources, or using single unit sources with uniform slip but with different areas? The explanation is not completely clear. Lack of use of earthquake scaling laws (ala "VAUS" sources in Davies et al. 2019) is a clear limitation if uniform slip is being used.

Addressed: We are not superposing unit sources. We have added a sentence to the supplementary methods to make that abundantly clear (line 71). For each magnitude step, we are using a variable amount of unit sources over which slip can be distributed. The length and width of the ruptures is constrained by the scaling relations of Strasser et al. (2010). The reviewer is correct that our approach is similar to the variable-area-uniform-slip (VAUS) approach in Davies et al. (2019) (lines 60-61 supplementary methods).

Description of how slip is distributed from the supplementary methods lines 80-84:

“For example, a $M_w 8.1$ can be composed of one, two, or three unit sources, translating into slips of $D \approx 8$ m (one unit source), $D \approx 4$ m (two unit sources), $D \approx 2.7$ m (three unit sources). Thus, for each earthquake magnitude, slip can be concentrated over a variable amount of unit sources to account for slip variability within each magnitude step. The minimum, average, and maximum slip and resultant deformation for each magnitude step is reported in Table S1.”

Line 111: You do not demonstrate anywhere that this is a robust estimate. A robust estimate would recreate observed tsunami variability. Please remove this statement. Fig S3 clearly shows that the method of Li et al. gives a broader distribution. Hence, you do not fully cover this uncertainty. Rather than stating that this is a "robust estimate", it should be discussed among with other limitations (such as on the next page)

Addressed: Please see reply to point 2 above and lines 102-111 in the main text.

Lines 132-135: I suggest removing this part. Tsunami frequency for high magnitude earthquakes are strongly determined by tapering and uncertainties, and this is not considered in this paper. Lines 233, 247, 250, 264: Please remove statements of earthquake probability. As substantiated above, this is misleading, but also not necessary for this paper (sensitivity to M_w alone is sufficient).

No change. As described above, we retain statements of relative earthquake rates.

References added

Grezio, A., et al., (2017). Probabilistic tsunami hazard analysis: Multiple sources and global applications. *Reviews of Geophysics*, 55(4), 1158-1198.

Davies, G. *et al.* A global probabilistic tsunami hazard assessment from earthquake sources. *Geol. Soc. Lond. Spec. Publ.* **456**, 219–244 (2018).

Kalligeris, N., Montoya, L., Ayca, A. & Lynett, P. An approach for estimating the largest probable tsunami from far-field subduction zone earthquakes. *Nat. Hazards* **89**, 233–253 (2017).

Reviewer #3 (Remarks to the Author):

New Title: Changing impacts of Alaska-Aleutian subduction zone tsunamis in California under future sea-level rise

Authors: Tina Dura et al.

Review by Debi Kilb, OK to give the authors my name.

This is the second time I am reviewing this article. I was impressed with the authors detailed response to the reviews, they carefully considered the reviewers comments and very clearly outlined changes and updates and carefully described why they disagreed with a few of the reviewers' suggestions.

When I first read this manuscript, my biggest concern was if uncertainties in the B-values were propagated through to the final results that the results would not remain robust. The authors examined this by assuming ± 0.1 uncertainties in B-values and determined that tests using the wider range of B-values produced results that did not change significantly. I appreciate the authors looking into this matter and I think this update makes the paper stronger.

A secondary concern of mine on the first reading of this manuscript was my confusion as to why the median value listed on line 151 of the original manuscript exceeded the maximum value listed in Table 1 in the original manuscript. The authors explained that the median value on line 151 referred to the medians of the MNTD distributions, not the medians of the sea-level rise projection. This clears up my confusion. If they have room, I suggest this distinction be added to the manuscript, but I don't feel all that strongly about the addition.

I like the new title.

I look forward to seeing this manuscript published in Nature Communications.

Dr. Debi Kilb
Scripps Institution of Oceanography

REVIEWER COMMENTS

Reviewer #1 (Remarks to the Author):

Review of revised paper entitled "Changing impacts of Alaska-Aleutian subduction zone tsunamis in California under future sea-level rise", by Tina Dura and co-authors

First, let me thank the authors for providing once again a thorough response and revision. The second issue raised is completely solved, while some more discussion is necessary for the first point.

I apologise if the authors felt that I tried to "move the goalposts" by bringing up a new subject in the second revision, this was not the intention. Rather, it was to raise concern related to the issue related to the uncertainty related to the M_w rates, which still isn't fully addressed in the paper. As a minimum, if the relative frequencies related to earthquake are retained, they must be accompanied with a discussion related to earthquake frequency uncertainty for the highest magnitudes. I realise that this should have been formulated better and elaborated more in the previous review. I try to explain my reasoning considering the authors rebuttal below, focussing only this remaining issue. I hope this will make the remaining revision easier for the authors.

Tsunami frequency and probability at the tail of the probability distribution is more uncertain than for the moderate magnitudes, but relative rates are presented without uncertainty in the main body of the paper, and this uncertainty is not discussed here. This holds for magnitudes where there are limited or no data (extrapolation):

Author rebuttal: However, the Alaska-Aleutian subduction zone has robust seismic catalog from M_5 - $M_{9.2}$ so using the b-value in this case to estimate of the relative recurrence of large-magnitude events is appropriate.

Response: The key here is figure Fig S5, which the authors uses to derive relative frequencies. To derive this curve, there is one event $>M_w 8.7$ (plots in the diagram as $M_w 9.1$, should this be $M_w 9.2$?). The uncertainty in the rates increases with M_w is likely substantial already beyond $M_w 8.7$. Based on this curve presented by the authors, we have to assume that the curve is extrapolated beyond $M_w 9.1$. The uncertainty related to MFD curve clearly increases with M_w . This also implies that the MFD curve potentially could take another shape in the tail of the distribution where M_w is largest, for instance as a tapered distribution. This uncertainty is not simply related only to the b-value only. Such uncertainties are not discussed here, and some discussion should enter in relation to the results presented on pages 13-14. In particular, this can affect the behaviour beyond the $M_w 9.1$ - 9.2 observation limit. Absolute uncertainty implies also that the relative rates are uncertain (you present some related to changing the b-values in the supplements), because you compare the frequencies of different M_w s along the MFD curve by taking ratios of the absolute values.

On pages 13-14 you compare the following relative frequencies: $M_w 8.0$ vs $M_w 8.7$, $M_w 8.8$ vs $M_w 9.1$, $M_w 9.0$ vs $M_w 9.4$, and $M_w 8.4$ vs $M_w 9.4$. While the $M_w 8.0$ vs $M_w 8.7$ comparison is well within the observational record, at least the last two or possibly three comparisons imply substantial uncertainties. I would suggest that these increasing uncertainties with magnitude are discussed more in depth. Simply giving a factor without any discussion related to an uncertainty that likely increases more for the highest magnitudes than for the smallest, is not sufficient.

Reviewer #1 (Remarks to the Author):

Review of revised paper entitled "Changing impacts of Alaska-Aleutian subduction zone tsunamis in California under future sea-level rise", by Tina Dura and co-authors

VIRGINIA POLYTECHNIC INSTITUTE AND STATE UNIVERSITY

An equal opportunity, affirmative action institution

First, let me thank the authors for providing once again a thorough response and revision. The second issue raised is completely solved, while some more discussion is necessary for the first point.

I apologise if the authors felt that I tried to "move the goalposts" by bringing up a new subject in the second revision, this was not the intension. Rather, it was to raise concern related to the issue related to the uncertainty related the Mw rates, which still isn't fully addressed in the paper. As a minimum, if the relative frequencies related to earthquake are retained, they must be accompanied with a discussion related to earthquake frequency uncertainty for the highest magnitudes. I realise that this should have been formulated better and elaborated more in the previous review. I try to explain my reasoning considering the authors rebuttal below, focussing only this remaining issue. I hope this will make the remaining revision easier for the authors.

Tsunami frequency and probability at the tail of the probability distribution is more uncertain than for the moderate magnitudes, but relative rates are presented without uncertainty in the main body of the paper, and this uncertainty is not discussed here. This holds for magnitudes where there are limited or no data (extrapolation):

Author rebuttal: However, the Alaska-Aleutian subduction zone has robust seismic catalog from M5-M9.2 so using the b-value in this case to estimate of the relative recurrence of large-magnitude events is appropriate.

Response: The key here is figure Fig S5, which the authors uses to derive relative frequencies. To derive this curve, there is one event $>Mw8.7$ (plots in the diagram as $Mw9.1$, should this be $Mw9.2?$).

Comment: The way the data is binned makes the point fall at 9.1 but it's the 9.2 earthquake.

The uncertainty in the rates increases with Mw is likely substantial already beyond $Mw8.7$. Based on this curve presented by the authors, we have to assume that the curve is extrapolated beyond $Mw9.1$. The uncertainty related to MFD curve clearly increases with Mw. This also implies that the MFD curve potentially could take another shape in the tail of the distribution where Mw is largest, for instance as a tapered distribution. This uncertainty is not simply related only to the b-value only. Such uncertainties are not discussed here, and some discussion should enter in relation to the results presented on pages 13-14. In particular, this can affect the behaviour beyond the Mw 9.1-9.2 observation limit. Absolute uncertainty implies also that the relative rates are uncertain (you present some related to changing the b-values in the supplements), because you compare the frequencies of different Mws along the MFD curve by taking ratios of the absolute values.

On pages 13-14 you compare the following relative frequencies: $Mw8.0$ vs $Mw8.7$, $Mw8.8$ vs $Mw9.1$, $Mw9.0$ vs $Mw9.4$, and $Mw8.4$ vs $Mw9.4$. While the $Mw8.0$ vs $Mw8.7$ comparison is well within the observational record, at least the last two or possibly three comparisons imply substantial uncertainties. I would suggest that these increasing uncertainties with magnitude are discussed more in depth. Simply giving a factor without any discussion related to an uncertainty that likely increases more for the highest magnitudes than for the smallest, is not sufficient.

Addressed: We have removed any reference to relative frequency of >M9.2 earthquakes from the manuscript.

We have edited the manuscript text that discusses the use of the earthquake catalog and b-values to reflect this (lines 122-125; pasted in below).

Lines 122-125

“We used a declustered earthquake catalog (b-value= 0.75) to calculate the relative rates of M5-9.2 earthquakes along the Alaska-Aleutian subduction zone (Table S2)^{39,40}. Due to the uncertainty associated with extrapolating the frequency of earthquakes not present in the earthquake catalog (>M9.2), we do not attempt to do so^{41,42}.”

We have removed the relative frequency of M9.4 earthquakes from page 14.

We have also removed the last column of Table S2, which had the relative frequency of M9.4 earthquakes.